# Renal interstitial cells promote nephron regeneration by secreting prostaglandin E2

Xiaoliang Liu[1†], Ting Yu[2†], Xiaoqin Tan[1], Daqing Jin[3], Wenmin Yang[1], Jiangping Zhang[1], Lu Dai[1], Zhongwei He[1], Dongliang Li[3], Yunfeng Zhang[1], Shuyi Liao[1], Jinghong Zhao[1]*, Tao P Zhong[3]*, Chi Liu[1]*

[1]Department of Nephrology, the Key Laboratory for the Prevention and Treatment of Chronic Kidney Disease of Chongqing, Chongqing Clinical Research Center of Kidney and Urology Diseases, Xinqiao Hospital, Army Medical University, Chongqing, China; [2]Department of Respiratory Medicine, Xinqiao Hospital, Army Medical University, Chongqing, China; [3]Shanghai Key Laboratory of Regulatory Biology, Shanghai Frontiers Science Center of Genome Editing and Cell Therapy, East China Normal University, School of Life Sciences, Shanghai, China

*For correspondence:
zhaojh@tmmu.edu.cn (JZ);
tzhong@bio.ecnu.edu.cn (TPZ);
chiliu@tmmu.edu.cn (CL)

†These authors contributed equally to this work

Competing interest: The authors declare that no competing interests exist.

**Abstract** In organ regeneration, progenitor and stem cells reside in their native microenvironment, which provides dynamic physical and chemical cues essential to their survival, proliferation, and differentiation. However, the types of cells that form the native microenvironment for renal progenitor cells (RPCs) have not been clarified. Here, single-cell sequencing of zebrafish kidney reveals *fabp10a* as a principal marker of renal interstitial cells (RICs), which can be specifically labeled by GFP under the control of *fabp10a* promoter in the *fabp10a:GFP* transgenic zebrafish. During nephron regeneration, the formation of nephrons is supported by RICs that form a network to wrap the RPC aggregates. RICs that are in close contact with RPC aggregates express cyclooxygenase 2 (Cox2) and secrete prostaglandin E2 (PGE2). Inhibiting PGE2 production prevents nephrogenesis by reducing the proliferation of RPCs. PGE2 cooperates with Wnt4a to promote nephron maturation by regulating β-catenin stability of RPC aggregates. Overall, these findings indicate that RICs provide a necessary microenvironment for rapid nephrogenesis during nephron regeneration.

## Editor's evaluation

This fundamental work substantially advances our understanding of the kidney interstitium and how it influences kidney development focusing on zebrafish as a model organism. The evidence supporting the conclusions is compelling, using single-cell analysis combined with in vivo zebrafish studies to mechanistically explore the functional importance of the discovery. The work will be of broad interest to cell and developmental biologists as well as the kidney community.

## Introduction

Acute kidney injury (AKI) is the most common cause of organ dysfunction in critically ill adults with an incidence of around 34% and carries an observed in-hospital mortality as high as 62% (*Doyle and Forni, 2016*). One of the main reasons is that mammals can partly repair their nephrons (the functional units of the kidney), but cannot regenerate new ones. By contrast, zebrafish can regenerate a large number of nephrons rapidly after kidney injury to compensate for lost renal function (*Diep et al., 2011*). During nephron regeneration, *lhx1a*-positive (*lhx1a*+) RPCs congregate to form cell

aggregates, then proliferate rapidly and differentiate into renal vesicles (RVs), which finally develop into mature nephrons (*Diep et al., 2011*). The native microenvironment in which progenitor and stem cells reside provides dynamic physical and chemical cues for their survival, proliferation, and function (*Lee et al., 2019*). However, the types of cells that form the native microenvironment of RPCs have not been elucidated.

RICs are specialized fibroblast-like cells localized mainly between the tubular and vascular structures, and thought to be responsible for the production of collagenous and noncollagenous extracellular matrices after kidney injury (*Whiting et al., 1999*). Under normal physiological conditions, RICs are the major sites of cyclooxygenase (Cox) expression and Cox-derived prostanoid synthesis (*Zhang et al., 2018*). Prostanoids regulate various aspects of renal physiology, including tubular transport and hemodynamics (*Nasrallah et al., 2007*). Numerous studies have reported the diverse and crucial roles of RICs during renal development, homeostasis, and disease (*Whiting et al., 1999*; *Nasrallah et al., 2007*; *Zhang et al., 2018*). However, the roles of RICs during nephron regeneration remain undefined.

PGE2 is one of the most typical lipid mediators produced from arachidonic acid (AA) by Cox enzymes (*Park et al., 2006*) and has been demonstrated to play a crucial role in the development of pronephros (*Chambers et al., 2020*; *Marra et al., 2019*; *Poureetezadi et al., 2016*). Elevation of PGE2 can restrict the formation of the distal nephron segment and expand a proximal segment lineage in zebrafish (*Poureetezadi et al., 2016*). PGE2 also regulates the development of renal multiciliated cells, in which embryos deficient in PGE2 form fewer renal multiciliated cell progenitors and develop more transporter cells (*Marra et al., 2019*). However, whether PGE2 affects nephrogenesis during regeneration of the zebrafish mesonephros has not been investigated.

In this study, we have employed single-cell sequencing in zebrafish kidney and have revealed that *fabp10a* is a marker of RICs. RICs can be specifically labeled by GFP in the *fabp10a:GFP* transgenic kidney. The formation of nephrons is closely related to Cox2a-expressed RICs, which are in close contact with RPC aggregates during nephron regeneration. Inhibiting PGE2 production suppresses nephrogenesis. Furthermore, PGE2 interacts with the Wnt signaling pathway by regulating β-catenin stability in RPC aggregates and can rescue the nephron regeneration defect in *wnt4a* mutants. These findings illustrate that RICs secrete PGE2 to promote nephron regeneration by PGE2/Wnt interaction.

## Results

### Single-cell messenger RNA sequencing reveals RICs

Understanding the function of an organ requires the characterization of its cell types (*Panina et al., 2020*). To illustrate the functions of the various cell types in the zebrafish kidney, we sequenced kidney cells by 10X Genomics single-cell RNA sequencing. Six randomly selected 6-month-old zebrafish kidneys were used to obtain about 7147 cells for sequencing. 3902 cells passed the quality control and were incorporated into analyses. Our initial analysis of kidney samples identified distinct clusters comprising 12 different cell types (*Figure 1A*). Further, we identified unique gene expression signatures that defined each of these cell clusters. The identities of 11 clusters were readily assigned based on expression of known markers or previous sequencing data (*Tang et al., 2017*; *Diep et al., 2011*). For example, two cell clusters were composed of epithelial cell types that are highly conserved in mammals, including the distal tubule epithelial cells defined by expression of *solute carrier family 12 member 3 (slc12a3)*, *ATPase Na$^+$/K$^+$ transporting subunit alpha 1a, tandem duplicate 4 (atp1a1a.4)*, and *ATPase Na$^+$/K$^+$ transporting subunit beta 1*a (*atp1b1a*), as well as the proximal tubule epithelial cells defined by the expression of *solute carrier family 22 member 2 (slc22a2)*, *paired box 2a (pax2a)*, and *aminoacylase 1 (acy1)* (*Tang et al., 2017*). We also identified hematopoietic stem cells, erythrocytes, T cells, neutrophils, macrophages, and vascular endothelial cells (*Figure 1A and B*, *Figure 1—source data 1*).

Interestingly, we found a new cell type (cluster 12). By comparing our data with the single-cell RNA sequencing data of human and mouse kidneys, it was found that this cluster of cells expressed human or mouse RICs signature genes, including *collagen type 1 alpha 1b (col1a1b)*, *collagen type 1 alpha 2 (col1a2)*, *decorin (dcn)*, and *elastin microfibril interfacer 1a (emilin1a)* (*Young et al., 2018*; *Stewart et al., 2019*; *Ransick et al., 2019*; *Figure 1B*, *Figure 1—figure supplement 1A–C*). Therefore, we speculated that these cells were RICs. We next performed gene ontology (GO) analysis of all genes in this cell cluster and found that a significant percentage of genes corresponded to the

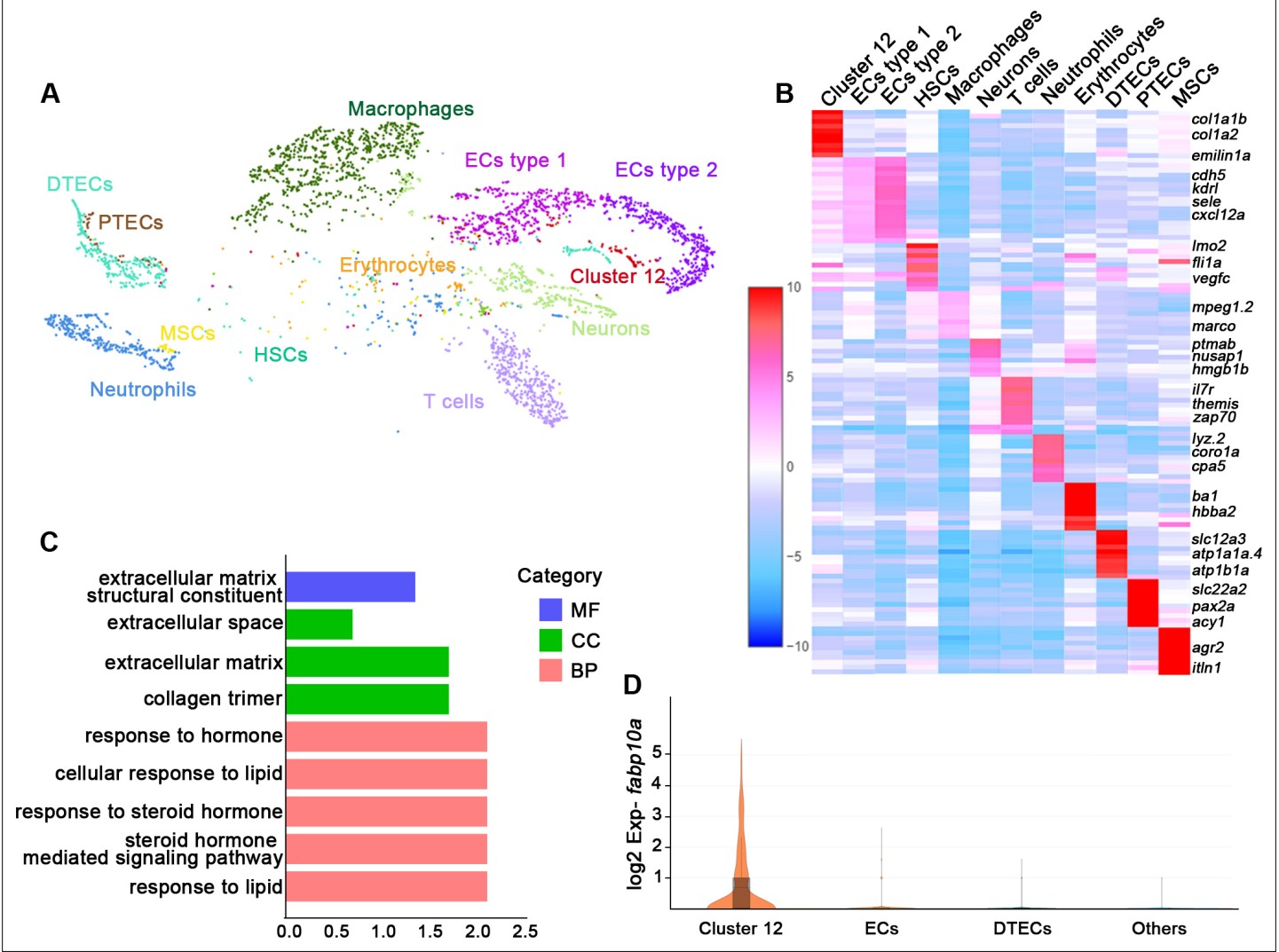

**Figure 1.** Single messenger RNA sequencing of kidney cells. (**A**) A tSNE plot showing clustering of all kidney cells after alignment using the Seurat package. Cells clustered with resolution 0.6. (**B**) Heat map showing relative log-expression of the top 2 or 3 marker genes for each cell cluster in **A**. (**C**) GO analysis of differentially expressed genes of cluster 12. A false discovery rate <0.05 was considered to indicate significant enrichment. BP, biological process; CC, cellular component; MF, molecular function. (**D**) Expression analysis of *fabp10a* in all clusters, showing that *fabp10a* was specifically expressed in cluster 12. DTECs, distal tubule epithelial cells; PTECs, proximal tubule epithelial cells; ECs, endothelial cells; HSCs, hematopoietic stem cells; MSCs, mucin-secreting cells; Exp, expression.

The online version of this article includes the following source data and figure supplement(s) for figure 1:

**Source data 1.** Gene expression of each identified cell population in zebrafish kidney single-cell RNA sequencing data.

**Figure supplement 1.** Expression analysis of interstitial cell marker genes in all kidney cell clusters.

extracellular matrix, the extracellular proteins that are not attached to the cell surface, or response to steroid hormones. Furthermore, this cluster of cells was also enriched in genes that respond to lipids (*Figure 1C*), which are richly contained by mammalian RICs (*Hao and Breyer, 2007*). Notably, we found that *fatty acid-binding protein 10a* (*fabp10a*), which plays a pivotal role in intracellular binding and trafficking of long-chain fatty acids (*Her et al., 2003*), was specifically expressed in cell cluster 12 but not in other type cells (*Figure 1D*). Taken together, we inferred *fabp10a* as a specific marker of this cell cluster.

### RICs are specifically marked by *fabp10a:GFP* in transgenic zebrafish

*Tg(fabp10a:GFP)* has been reported to specifically label the liver during early embryonic development (*Her et al., 2003*). To observe the expression of *fabp10a* in the kidney, we crossed *Tg(fabp10a:GFP)*

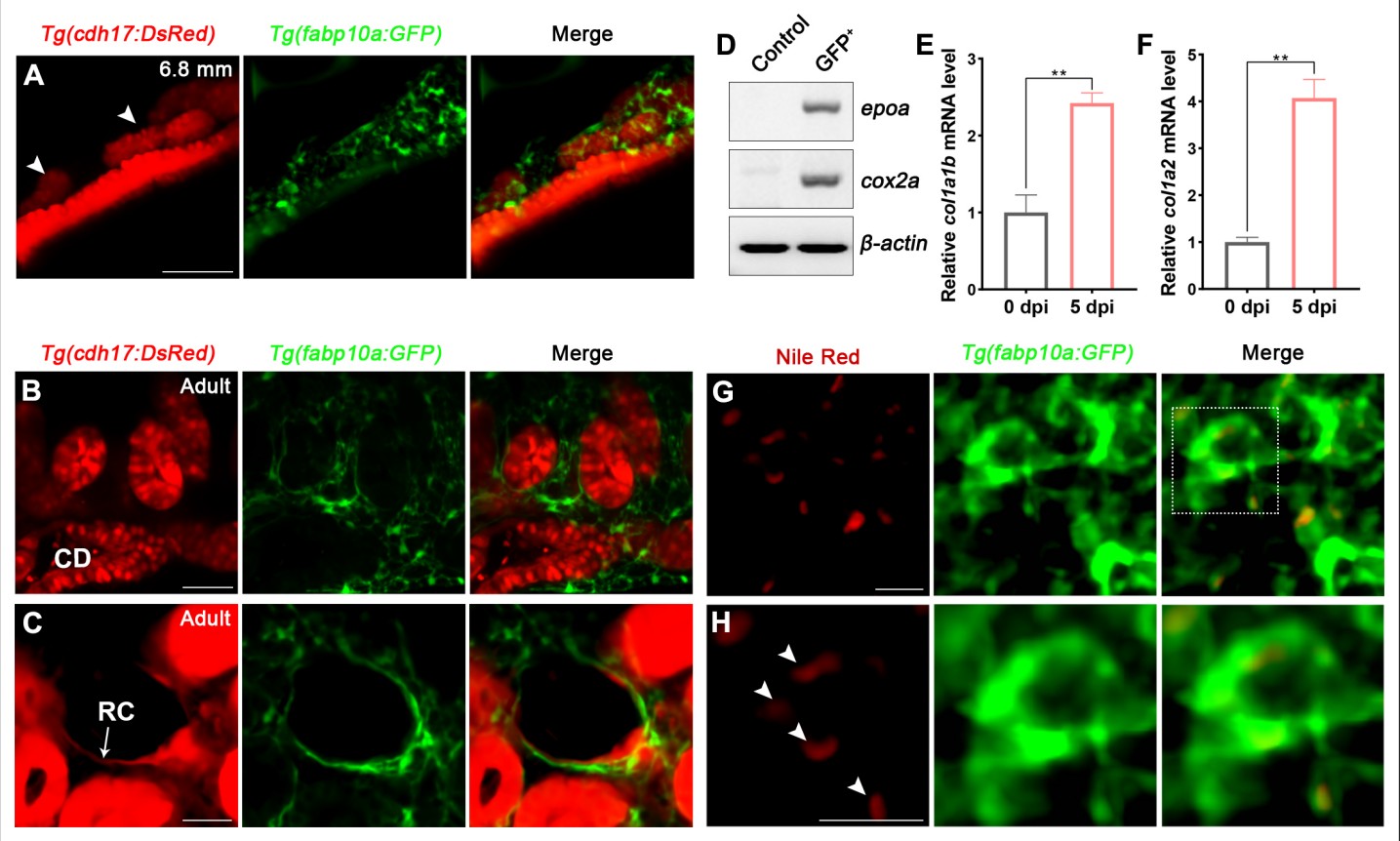

**Figure 2.** Renal interstitial cells (RICs) are specifically labeled by *fabp10a:GFP* in transgenic zebrafish. (**A**) Localization of *Tg(fabp10a:GFP)* labeled-RICs in 6.8 mm stage juvenile zebrafish. *Tg(cdh17:DsRed)* labeled renal tubules (arrowheads, new mesonephric branches; n = 6). (**B, C**) In adult zebrafish kidney, *Tg(fabp10a:GFP)* marks RICs (n = 3), while *Tg(cdh17:DsRed)* labels CD and also renal capsule (RC). *Tg(fabp10a:GFP)* cells form a network to tightly wrap kidney tubules (**B**) and capsules (**C**). CD, collecting duct; RC, renal capsule. (**D**) RT-PCR analysis of the expression of *epoa* and *cox2a*. *β-actin* was used as a sample control. GFP[+] indicates cells with only GFP fluorescence; control indicates all cells except GFP[+]/DsRed[-] cells. (**E, F**) *col1a1b* and *col1a2* mRNA levels in *Tg(fabp10a:GFP)*-labeled GFP[+]/DsRed[-] cells were quantified by qPCR. Both were significantly increased at 5 dpi (n = 3). Both genes were normalized to the mean expression level at 0 dpi, which was set to 1. **$p < 0.01$ by one-way ANOVA. (**G**) Nile red staining section of *Tg(fabp10a:GFP)* zebrafish kidney showing that *Tg(fabp10a:GFP)*-labeled cells contained plentiful lipid droplets. (**H**) Higher-magnification image of the boxed area showed in (**G**) (arrowheads, lipid droplets). n = 3. Scale bar in (**A–C**), 100 μm; (**G, H**), 20 μm.

The online version of this article includes the following source data and figure supplement(s) for figure 2:

**Source data 1.** Original gel files of *Figure 2D* and *Figure 2—figure supplement 1B*.

**Source data 2.** Numerical data for *Figure 2E and F*.

**Figure supplement 1.** Renal interstitial cells (RICs) can be labeled by *Tg(fabp10a:GFP)* line.

with *Tg(cdh17:DsRed)*, which is a transgenic line specifically labeled nephron epithelial cells, and found that *Tg(fabp10a:GFP)* labeled not only the liver and spinal cord but also a part of proximal straight tubule (PST) and distal tubule (DT) starting at 4 days post-fertilization (4 dpf) (*Figure 2—figure supplement 1A*). Furthermore, a large number of *Tg(fabp10a:GFP)* labeled cells were observed to distribute around the area where the mesonephric tubules appeared after 15 dpf (*Figure 2A*, *Figure 2—figure supplement 1A*). These cells were fusiform with irregular polygon shapes, and interweaved between cells to form a network (*Figure 2A*), similar to the same cell shape as mammalian RICs (*Whiting et al., 1999*; *Zhang et al., 2018*).

To further identify these cells, we analyzed adult kidney sections of the *Tg(fapb10a:GFP;cdh17:DsRed)* double transgenic line. A large number of GFP[+] cells were attached to the renal tubules and collecting ducts, or distributed in the renal interstitium (*Figure 2B*). Moreover, these cells also tightly surrounded the *cdh17*-labeled renal capsules (*Figure 2C*). As in juvenile zebrafish, these cells had multitudinous shapes and formed a network (*Figure 2A–C*). The distribution and morphology of

these cells were similar to those of the RICs in the mammalian metanephros (*Whiting et al., 1999*; *Zhang et al., 2018*). Next, we sorted the adult kidney cells of the *Tg(fapb10a:GFP;cdh17:DsRed)* line by flow cytometry (FACS). Two groups of cells were collected. One group of cells only expressed green fluorescence (GFP+/DsRed-), and the control group included all of the other cells, except GFP+/DsRed- cells. We detected the *fabp10a* expression in GFP+/DsRed- cells but not in control cells by semi-quantitative reverse transcription-polymerase chain reaction (RT-PCR) (*Figure 2—figure supplement 1B*), indicating that GFP+/DsRed- cells were indeed *fabp10a* positive. Furthermore, we detected the enrichment of *erythropoietin (epo)* gene, a marker of mammalian RICs (*Kobayashi et al., 2016*; *Souma et al., 2015*), in GFP+/DsRed- cells, but not in control cells (*Figure 2D*). Cox2 is also considered to be a marker of mammalian RICs (*Zhang et al., 2018*). We found that *cox2a*, the *Cox2*-orthologous gene in zebrafish, was also highly enriched in GFP+/DsRed- cells (*Figure 2D*).

RICs have been reported to differentiate into fibroblasts when AKI occurs in mammals (*Baues et al., 2020*). We intraperitoneally injected gentamicin (Gent; 2.7 µg/µL, 20 µL per fish), an established nephrotoxin, into zebrafish to induce AKI (*Diep et al., 2011*; *Augusto et al., 1996*) and then collected the GFP+/DsRed- kidney cells of *Tg(fapb10a:GFP;cdh17:DsRed)* zebrafish at 5 days post-injury (5 dpi) by FACS. Compared with the uninjured kidneys, mRNA levels of fibroblast markers *col1a1b* and *col1a2* were increased significantly in GFP+/DsRed- cells at 5 dpi according to quantitative PCR (qPCR) analysis (*Figure 2E and F*). Because mammalian RICs are lipid-rich cells (*Hao and Breyer, 2007*), Nile red, a fluorescent dye that specifically stains lipids, was used to detect lipid droplets in adult renal tissues. Many lipid droplets were detected in the cytoplasm of most *Tg(fabp10a:GFP)* labeled cells (*Figure 2G and H*). Collectively, these findings demonstrated that this cluster of cells had a high degree of similarity and conservation as mammalian RICs based on cell location, morphology, and molecular or biochemical characteristics. Thus, we defined the cluster of *fabp10a+* cells as the zebrafish RICs.

## Kidney injury promotes synthesis and secretion of PGE2 by RICs

Kidney injury in adult zebrafish induces the synchronous formation of many new nephrons marked by *lhx1a* (*Kamei et al., 2019*; *Diep et al., 2011*). In the *Tg(fapb10a:GFP;lhx1a:DsRed)* injured kidneys, we found that the *lhx1a+* RPC aggregates were wrapped around by the RICs that closely interacted with *lhx1a+* cells (*Figure 3B*, *Figure 3—figure supplement 1A*). Additionally, EdU assay showed that RICs and *lhx1a+* RPCs proliferated simultaneously after AKI (*Figure 3—figure supplement 1B*), suggesting that RICs participate in the regeneration of nephrons.

To further validate the interaction between RICs and RPCs, we assessed the real-time interaction in the *Tg(fabp10a:GFP;cdh17:DsRed)* or *Tg(fabp10a:GFP; lhx1a:DsRed)* juvenile zebrafish using time-lapse fluorescence microscopy imaging technique (*Figure 3—figure supplement 2A–F*, *Figure 3—video 1*). During the development of mesonephros from 10 dpf to 16 dpf, RPCs occurred with the pronephric tubules at the 4.6 mm stage (body length; *Figure 3—figure supplement 2A*) and RICs were detected around the RPC at the 5 mm stage. At this time, the RPCs congregate to form the first cell aggregate (*Figure 3—figure supplement 2B and D*). During the differentiation of this cell aggregate, more RICs appeared and covered the cell aggregate as a cell network. The RPC aggregate then elongated to form the first mesonephric branch at the 5.3 mm stage, and large numbers of RICs appeared in the direction of elongation (*Figure 3—figure supplement 2C and E*). Further, the mesonephric tubule elongated and formed the nascent nephron at the 5.8 mm stage, and RICs formed a network that completely wrapped the entire nephron (*Figure 3—figure supplement 2F*, *Figure 3—video 1*). As the body grew, other nascent nephrons appeared in the pronephric tubules. RICs repeat the above process to produce new cell networks to wrap nascent nephrons (*Figure 2A*). This finding demonstrates that the RIC network is a basic element of nephron structure. Furthermore, we also determined the proliferation of RICs during this process, and found a large number of RICs were EdU+, as in regeneration stage (*Figure 3—figure supplement 2G*), indicating that cell proliferation during nephrogenesis is a source of RICs. The above results indicate that RICs are closely related to nephrogenesis.

In order to analyze the molecular mechanism of nephron regeneration, we performed the renal transcriptome analyses using injured kidney tissues at 1, 3, 5, and 7 dpi (*Figure 3—source data 1*). The transcriptome data showed that the expression of *cox2a*, a marker gene of RICs, was increased significantly after AKI compared to uninjured tissues. Further, we confirmed this result by RT-PCR and found that *cox2a* mRNA expression increased at 1 dpi, reached its highest level at 5 dpi, and returned

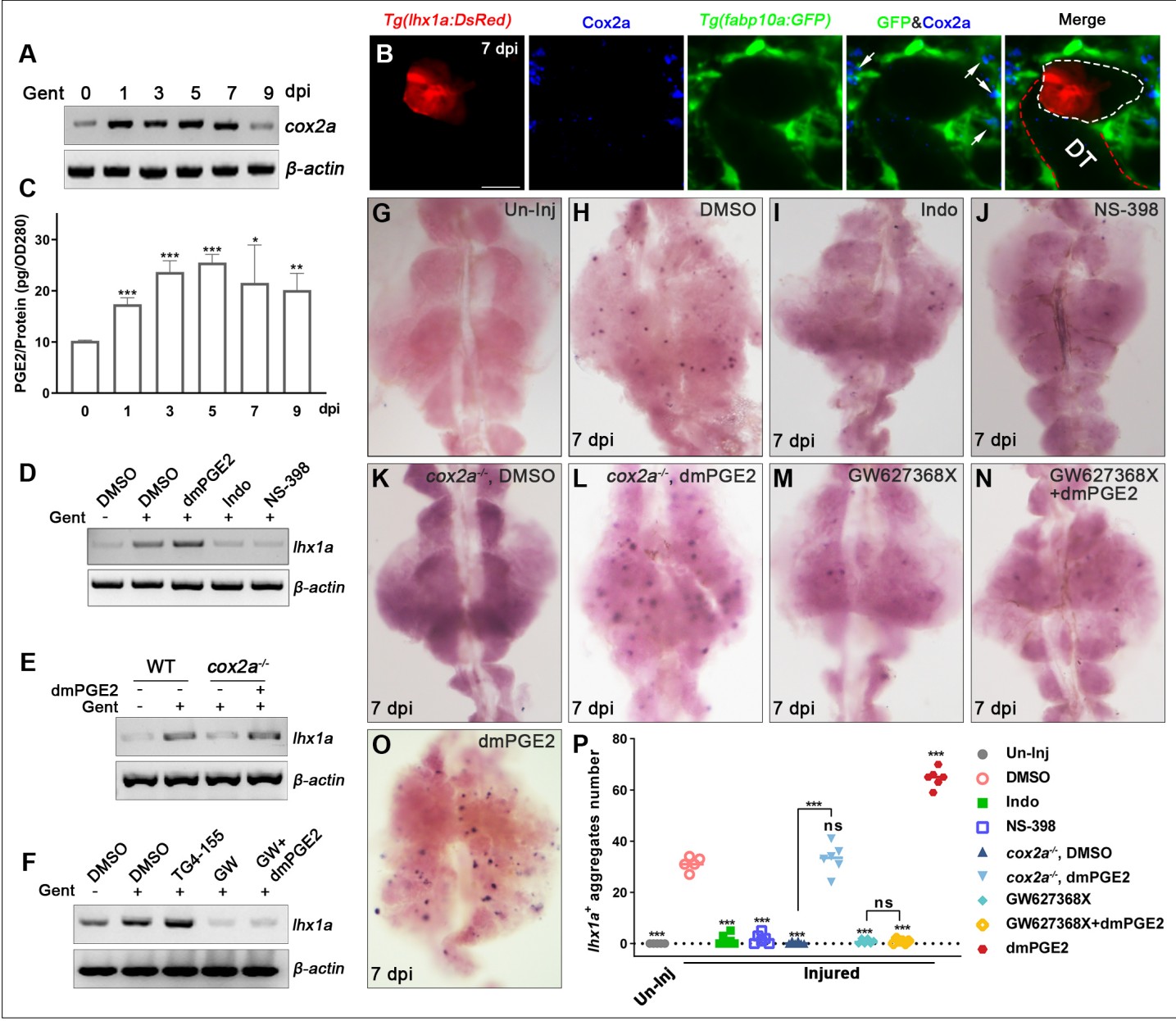

**Figure 3.** Renal interstitial cells (RICs) promote nephron regeneration by secreting PGE2. (**A**) *cox2a* mRNA levels were evaluated by RT-PCR at 0, 1, 3, 5, 7, and 9 dpi. *β-actin* was used as a sample control. The expression of *cox2a* was upregulated after acute kidney injury (AKI) and reached its peak level at 5 dpi (n = 3). (**B**) Immunofluorescence staining of Cox2a in *Tg(fabp10a:GFP;lhx1a:DsRed)* zebrafish kidneys at 7 dpi. RICs around *lhx1a*[+] cell aggregates highly expressed Cox2a (arrows; DT, distal tubule, red outline; *lhx1a*[+] cell aggregates, white outline; n = 4) Scale bar, 50 μm. (**C**) PGE2 levels were assessed at 0, 1, 3, 5, 7, and 9 dpi using PGE2 ELISA kits. PGE2 levels increased after AKI and reached their peak at 5 dpi (n = 3). Data were analyzed by ANOVA, *p<0.05, **p<0.01, ***p<0.001 vs 0 dpi. (**D**) *lhx1a* mRNA levels were evaluated by RT-PCR at 7 dpi. (**E**) The *lhx1a* mRNA levels in *cox2a*[-/-] and WT zebrafish kidneys were assessed by RT-PCR at 7 dpi. (**F**) The *lhx1a* mRNA levels were determined by RT-PCR at 7 dpi after injection of EP2 inhibitor TG4-155 or EP4 inhibitor GW627368X (GW). *β-actin* was used as a sample control in (**D–F**). (**G–O**) *lhx1a* WISH showing the trunk kidney region at 7 dpi (n = 5–7). (**G**) *lhx1a*[+] cell aggregates could not be detected in un-injured (Un-Inj) kidneys. (**H**) Injury induced the formation of *lhx1a*[+] cell aggregates. Indo (**I**), NS-398 (**J**), or Cox2a deficiency (**K**) inhibited the formation of *lhx1a*[+] cell aggregates. dmPGE2 could rescue the effect of Cox2a deficiency (**L**). GW627368X inhibited the formation of *lhx1a*[+] cell aggregates (**M**), and dmPGE2 could not rescue the defect (**N**). dmPGE2 could promote the formation of *lhx1a*[+] cell aggregates (**O**). (**P**) *lhx1a*[+] cell aggregates in uninjured and injured kidneys that were treated with DMSO, COX inhibitors or dmPGE2 were calculated using ImageJ. n = 5–7 in each condition. Data were analyzed by ANOVA, *p<0.05, **p<0.01, ***p<0.001.

The online version of this article includes the following video, source data, and figure supplement(s) for figure 3:

**Source data 1.** Differential analysis of gene expression in zebrafish nephron regeneration transcriptome.

*Figure 3 continued on next page*

*Figure 3 continued*

**Source data 2.** Original gel files of *Figure 3A, D and F*.

**Source data 3.** Numerical data for *Figure 3C and P* and *Figure 3—figure supplement 1D–F*.

**Figure supplement 1.** Kidney injury promotes renal interstitial cells (RICs) to synthesize and secrete PGE2.

**Figure supplement 2.** The interaction between renal interstitial cells (RICs) and renal progenitor cells (RPCs).

**Figure 3—video 1.** 3D video of renal interstitial cells (RICs) wrapping renal progenitor cell (RPC) aggregate corresponds to Figure 3—figure supplement 2F.

https://elifesciences.org/articles/81438/figures#fig3video1

to normal at 9 dpi (*Figure 3A*). Furthermore, immunofluorescence revealed that Cox2a was increased during nephron regeneration (*Figure 3—figure supplement 1C*). Interestingly, most of the Cox2a-positive RICs were in close contact with RPC aggregates (*Figure 3B*).

Cox2 is a rate-limiting enzyme in PGE2 synthesis (*Nørregaard et al., 2015*; *Poureetezadi et al., 2016*). We examined PGE2 levels in kidneys using immunosorbent (ELISA) assay and found that PGE2 was increased significantly during nephron regeneration. The PGE2 levels were about 69.7% higher in injured groups than uninjured control groups at 1 dpi and reached their maximum value at 5 dpi (*Figure 3C*). This pattern was consistent with the change in *cox2a* mRNA in the transcriptome analysis (*Figure 3A*). We next evaluated the expression of *cox1*, *cox2a*, and *cox2b*, the zebrafish *cox* gene family, by qPCR. The results showed that the expression of *cox2a* was increased (*Figure 3A*), whereas *cox2b* expression was reduced during nephron regeneration. However, *cox1* expression remained unaltered (*Figure 3—figure supplement 1D and E*). Altogether, these results indicate that AKI could induce RICs to specifically express *cox2a* that increases the synthesis and secretion of PGE2.

## Blockade of the Cox2a-mediated PGE2 synthesis leads to the defective nephron regeneration

Our results show that RICs, which surround RPC aggregates, highly expressed Cox2a and secreted PGE2, suggesting that PGE2 participates in nephron regeneration. To assess whether endogenous PGE2 participates in nephron regeneration after AKI, we intraperitoneally injected indomethacin (Indo, 200 µM, 10 µL per fish), an inhibitor of both Cox1 and Cox2, at 2, 4, and 6 dpi (*Poureetezadi et al., 2016*). Whole-mount in situ hybridization (WISH) showed that the number of *lhx1a*$^+$ cell aggregates was decreased at 7 dpi compared with the control group (*Figure 3D, G–I, P*). *lhx1a* expression was also reduced (*Figure 3D*). These results indicate that suppressing Cox activity restrains nephron regeneration. Given that the *cox2a* expression was significantly upregulated following AKI (*Figure 3A*), we infer that Cox2a has a decisive role in PGE2 synthesis during regeneration. To test this, N-(2-cyclohexyloxy-4-nitrophenyl) methane sulfonamide (NS-398, 140 µM, 10 µL per fish), a selective Cox2 inhibitor (*Marra et al., 2019*), was injected after AKI. As a result, *lhx1a* expression and the number of *lhx1a*$^+$ cell aggregates was decreased, as in the Indo-treated group (*Figure 3D, J and P*).

To further explore these findings, we used a genetic model of Cox2a deficiency to assess the effects of the Cox2a-PGE2 pathway on nephron regeneration. There were no obvious morphological differences between *cox2a*$^{-/-}$ and wild-type (WT) sibling fish (*Li et al., 2019*). We next induced AKI in *cox2a*$^{-/-}$ mutants and found that *lhx1a* mRNA and the number of *lhx1a*$^+$ cell aggregates in *cox2a*$^{-/-}$ kidneys were less than injured WT kidneys at 7 dpi (*Figure 3E, K and P*), indicating that Cox2a is required for nephron regeneration. We next measured PGE2 level in *cox2a*$^{-/-}$ kidneys at 5 dpi and found it was significantly lower than that in injured WT kidneys, while no significant differences were detectable between uninjured *cox2a*$^{-/-}$ and WT groups (*Figure 3—figure supplement 1F*). Furthermore, we injected the stable PGE2 analog dmPGE2 (600 µM, 10 µL per fish) into injured *cox2a*$^{-/-}$ kidneys (*Goessling et al., 2009*; *Marra et al., 2019*), and observed that the expression of *lhx1a* and the number of *lhx1a*$^+$ cell aggregates were restored to the levels in injured WT kidneys at 7 dpi (*Figure 3E, H, L and P*). To assess the effects of PGE2 on the induction in nephron regeneration, we injected dmPGE2 (600 µM, 10 µL per fish) into WT zebrafish following AKI and observed significant increases in both the expression of *lhx1a* and the number of *lhx1a*$^+$ cell aggregates, in comparison with that in DMSO-treated groups at 7 dpi (*Figure 3D, O and P*), indicating that PGE2 is sufficient for nephron regeneration.

## PGE2–EP4 signaling promotes the proliferation of RPCs during regeneration

During nephron regeneration, RPCs congregate to form cell aggregates, then proliferate rapidly and differentiate into RVs (*Diep et al., 2011*). Therefore, we tested the proliferation of RPCs by EdU incorporation assay and found that about 43.5% of *lhx1a*+ RPCs incorporated EdU in the process of proliferation. However, only about 5.5% of *lhx1a*+ RPCs displayed EdU+ nuclei during RPC aggregation (*Figure 4—figure supplement 1A*). Next, we observed that the RPCs in the cell aggregate differentiate into square epithelial cells from irregular mesenchymal cells (*Figure 4—figure supplement 1B and C*). Cadherins responsible for cell–cell adhesion were also increased in the aggregates (*Figure 4—figure supplement 1D and E*). Notably, the *lhx1a*+ RPC aggregates remained small in *cox2a*-/- mutants compared to WT at 5 dpi (*Figure 4—figure supplement 1F and G*), suggesting that PGE2 signaling mediates the proliferation of RPC aggregates.

To test whether PGE2 affects proliferation of RPCs during nephron regeneration, we assessed the proliferation of RPCs following AKI in adult zebrafish. First, we blocked PGE2 synthesis with Indo or NS-398 after AKI, and analyzed the EdU incorporation of DsRed+ RPCs in cell aggregates in *Tg(lhx1a:DsRed)* kidneys. The results showed that the number of RPCs in the cell aggregate was about 5–15 cells in the Indo- or NS-398-treated group at 5 dpi (*Figure 4—figure supplement 2C and E*). Among them, 10.9% of Indo-treated or 11.6% of NS-398-treated *lhx1a*+ RPCs exhibited EdU+ nuclei (*Figure 4—figure supplement 2A–G*). Moreover, these aggregates had no epithelial cell morphology as *cox2a*-/- mutants (*Figure 4—figure supplement 2A–F*). In contrast, the number of RPCs in DMSO-treated cell aggregates was increased to an average of 47 (*Figure 4A*, *Figure 4Figure 2A*), and about 39.9% of *lhx1a*+ RPCs displayed EdU+ nuclei (*Figure 4A, B and K*). Furthermore, we found that the number of RPCs in the cell aggregate in the *Tg(lhx1a:DsRed)/cox2a*-/- kidney was about 11 cells at 5 dpi, with about 10.3% of *lhx1a*+ RPCs being EdU-positive RPCs. Then, we injected dmPGE2 to *Tg(lhx1a:DsRed)/cox2a*-/- kidney following AKI. RPC numbers in the cell aggregate were restored to injured WT level at 5 dpi, with about 49.9% of *lhx1a*+ RPCs incorporating EdU (*Figure 4C–F and K*). These findings indicate that PGE2 secreted by RICs is required for the proliferation of RPCs during nephron regeneration.

PGE2 signals through four prostaglandin E receptors (EP1–4) in mammals (*Miyoshi et al., 2017*). To test which EP receptor mediates PGE2 signaling in RPCs, we obtained DsRed+ RPCs from the *Tg(lhx1a:DsRed)* transgenic line by FACS at 5 dpi. The qPCR results showed that *ep2a* and *ep4b* mRNAs but not *ep1*, *ep3*, or *ep4a* mRNAs were highly expressed in RPCs. Notably, the *ep4b* expression was about twice than the *ep2a* level (*Figure 4—figure supplement 3A*). Furthermore, *ep4b* was expressed in the RPC aggregates using WISH analyses (*Figure 4—figure supplement 3B*). We next injected a specific pharmacological inhibitor for EP2 (TG4-155, 400 μM, 10 μL per fish) or EP4 (GW627368X, GW, 200 μM, 10 μL per fish) into WT fish (*Baker and Van Der Kraak, 2019*). Only the EP4 inhibitor GW627368X blocked nephron regeneration (*Figure 3F, M and P*). Furthermore, injection of GW627368X restrained the proliferation of RPCs (only about 6.4% *hx1a*+ RPCs had EdU+ nuclei). Importantly, dmPGE2 did not rescue the GW627368X-induced inhibitory effect on RPC proliferation (*Figure 4G–K*) and nephron regeneration (*Figure 3F, N and P*). Taken together, these results indicate that PGE2 promotes nephrogenesis through the EP4 receptor during nephron regeneration.

## PGE2–EP4 signaling promotes nephron regeneration through the PGE2/Wnt interaction

Previous reports have shown that PGE2 signaling promotes the development of hematopoietic stem cells and liver regeneration through the PGE2/Wnt interaction (*Goessling et al., 2009*; *North et al., 2010*). To assess whether PGE2–EP4 signaling contributes to nephron regeneration through the Wnt signaling pathway, we tested Wnt activity at 7 dpi. As a result, the expression of *lef1*, a canonical marker of Wnt signaling activity (*Kamei et al., 2019*), was increased significantly in the injured kidneys compared with the uninjured kidneys. However, Wnt activity was significantly diminished in injured *cox2a*-/- kidneys at 7 dpi, but was rescued by the intraperitoneal dmPGE2 injection (*Figure 5A*). Furthermore, the expression of *lef1* was also significantly inhibited by injection of Indo, NS-398, or EP4-specific inhibitor GW627368X, but was significantly enhanced by injection of excessive dmPGE2 (*Figure 5—figure supplement 1A*). These findings reveal a link between PGE2 and Wnt activity.

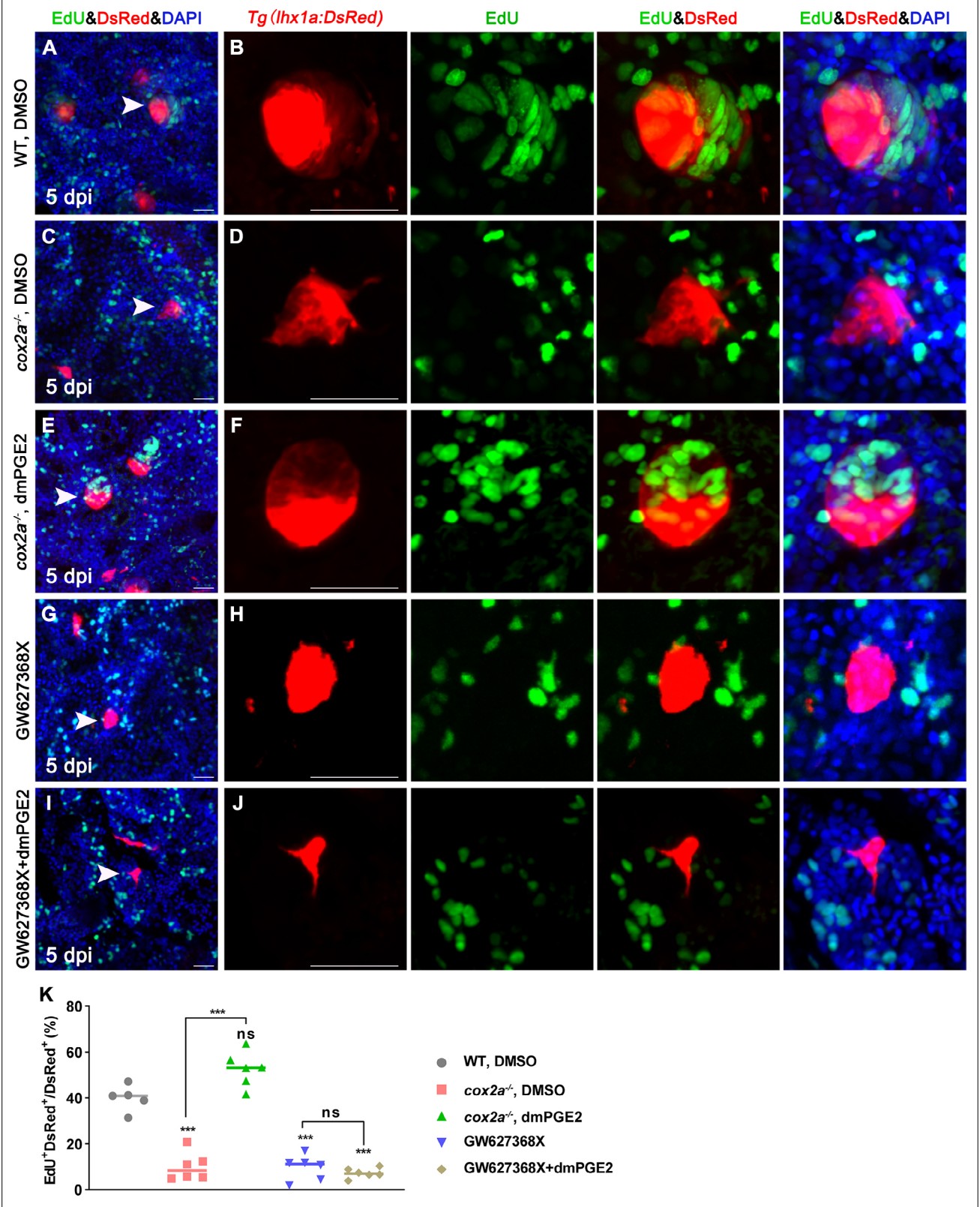

**Figure 4.** PGE2–EP4 signaling promotes the proliferation of renal progenitor cells (RPCs) during nephron regeneration. *Tg(lhx1a:DsRed)* fish were injected with EdU to label proliferating nuclei at 5 dpi and kidneys were harvested 3 hr later. (**A, B**) Gentamicin induced *lhx1a+* cell aggregates with proliferating EdU+ nuclei. (**C, D**) The proliferation of *lhx1a+* cell aggregates in *cox2a-/-* was significantly less than that in WT. (**E, F**) Intraperitoneal injection of dmPGE2 could promote the proliferation of *lhx1a+* cells in *cox2a-/-*. (**G, H**) GW627368X inhibited the proliferation of *lhx1a+* cells. (**I, J**) Injection of

*Figure 4 continued on next page*

*Figure 4 continued*

dmPGE2 could not rescue the GW627368X treatment. n = 5–7 in each condition. Scale bar in (**A, C, E, G, I**), 50 µm; (**B, D, F, H, J**), 100 µm. The images on the right showed a higher-magnification image (arrowheads). (**K**) Proliferation ratio of *lhx1a⁺* RPCs in (**A–J**) was calculated using ImageJ. n = 5–7 in each condition. Data were analyzed by ANOVA, ***p<0.001; ns, no significant difference.

The online version of this article includes the following source data and figure supplement(s) for figure 4:

**Source data 1.** Numerical data for *Figure 4K*, *Figure 4—figure supplement 1A*, and *Figure 4—figure supplement 3A*.

**Figure supplement 1.** The proliferation and differentiation of renal progenitor cells (RPCs) during regeneration.

**Figure supplement 2.** PGE2–EP4 signaling promotes the proliferation of renal progenitor cells (RPCs).

**Figure supplement 3.** PGE2 receptors expression in *lhx1a⁺* renal progenitor cell (RPC) aggregates.

Genetic studies in mice reported that Wnt/β-catenin signaling plays various roles during kidney development (*Pulkkinen et al., 2008*). Wnt4, expressed in the nephrogenic cap mesenchyme, induces the condensation of RPCs by mesenchymal to epithelial transition (MET) and also promotes the formation of RVs (*Carroll et al., 2005*; *Kispert et al., 1998*; *Stark et al., 1994*). We sorted *lhx1a⁺* cells by FACS using *Tg(lhx1a:DsRed)* kidneys at 5 dpi and found that *wnt4a*, the *Wnt4*-orthologous gene in zebrafish, was expressed in RPCs (*Figure 5B*). RT-PCR results indicated that *wnt4a* transcripts were increased at 5 dpi compared with uninjured kidneys (*Figure 5C*), indicative of the involvement of Wnt4a in the regeneration of nephrons. Next, we injured *wnt4a⁻/⁻* kidneys and discovered that both *lhx1a* expression and RPC aggregate number were significantly less than that in injured WT kidneys at 7 dpi (*Figure 5D–G*). Interestingly, the influence of Wnt4a deficiency was rescued by injection of dmPGE2 (*Figure 5D–I*). Furthermore, we detected that only about 4.6% of *lhx1a⁺* RPCs showed EdU incorporation in *wnt4a⁻/⁻* kidneys at 5 dpi (*Figure 5J–M and P*). After dmPGE2 injection, the effects of Wnt4a deficiency on nephron regeneration were eliminated and 42.8% of the *lhx1a⁺* RPCs exhibited EdU⁺ nuclei. No significant differences were detected in WT kidneys at 5 dpi (*Figure 5J–P*). Altogether, these findings suggest that RIC-secreted PGE2 cooperates with RPC-secreted Wnt4a to regulate nephron regeneration.

## PGE2 regulates the stability of β-catenin in regenerating RPCs through PKA

In the absence of Wnt ligand, intracellular β-catenin can be degraded by the β-catenin destruction complex formed by Axin, APC, GSK3β, and CK1 (*Stamos and Weis, 2013*; *Cadigan and Waterman, 2012*; *MacDonald and He, 2012*; *Ha et al., 2004*). To assess whether PGE2 regulates the Wnt signaling pathway through β-catenin destruction complex, we injected a Wnt pathway inhibitor, XAV939, which stimulates β-catenin degradation by stabilizing Axin (*Hofsteen et al., 2018*). It has been shown that XAV939 can inhibit nephron regeneration (*Kamei et al., 2019*). We found that the XAV939-mediated nephron regeneration defect could be rescued by the injection of dmPGE2 (*Figure 6A, D–G, L*). Similarly, the administration of ICRT 14 or ICG-001, the Wnt inhibitor that perturbs the interaction between β-catenin and TCF, impedes the nephron regeneration, which was neither rescued by the dmPGE2 injection (*Figure 6B–E, H–L*). These findings indicate that PGE2 acts downstream of the β-catenin destruction complex but upstream of the nuclear transcriptional interaction.

Next, we tested the β-catenin protein level in the RPC aggregates following AKI by immunofluorescence and observed the enrichment of β-catenin during RPC aggregation or proliferation (*Figure 6M, N and W*). On the contrary, β-catenin was hardly detected in the RPC aggregates in the injured *cox2a⁻/⁻* mutant kidney (*Figure 6O and W*). The administration of Indo-, NS-398-, or GW627368X of injured WT kidneys also decreased β-catenin in the regenerating RPC aggregates (*Figure 6—figure supplement 1A–D and H*). However, the injection of dmPGE2 into the *cox2a⁻/⁻* kidneys rescued the loss of β-catenin in the *lhx1a⁺* RPC aggregates following AKI (*Figure 6P, Q and W*). Furthermore, we found that β-catenin was less in *wnt4a⁻/⁻* RPC aggregates than WT aggregates at 5 dpi and that injection of dmPGE2 restored partially the β-catenin level in *wnt4a⁻/⁻* RPC aggregates (*Figure 6R–T and W*). To more clearly demonstrate the subcellular location of β-catenin, we sorted out the RPCs and performed β-catenin immunofluorescence. We observed that nuclear β-catein was significantly reduced in *cox2a⁻/⁻* or *wnt4a⁻/⁻* RPCs compared to WT RPCs, and dmPGE2 treatment can restored the reduced β-catein in these mutants (*Figure 6—figure supplement 2A–F*). To further prove that the change of nuclear β-catenin level affects the Wnt signal activity, we used fluorescence in situ hybridization (FISH) to

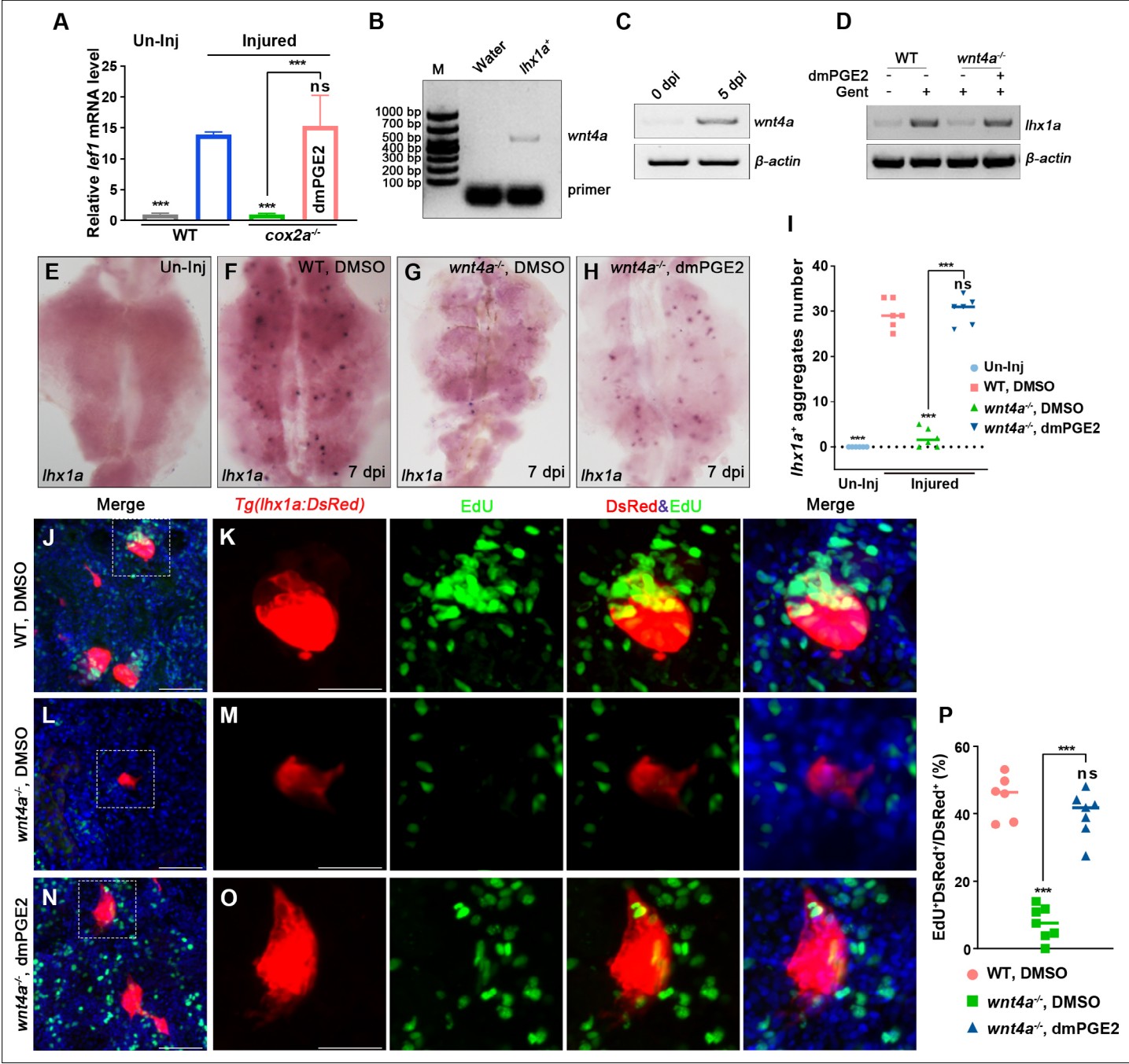

**Figure 5.** PGE2–EP4 signaling promotes nephron regeneration through PGE2/Wnt interaction. (**A**) qPCR relative quantification of *lef1* mRNA in kidney tissue of WT and *cox2a⁻/⁻* harvested at 7 dpi. Gene was normalized to the mean expression level in uninjured (Un-Inj) kidneys, which was set to 1. **p<0.01, ***p<0.001; ns, no significant difference (n = 3). (**B**) FACS-related RT-PCR analysis of *wnt4a* expression in *Tg(lhx1a:DsRed)* labeled RPCs at 5 dpi, and *wnt4a* was expressed in these cells. (**C**) The *wnt4a* mRNA levels were assessed by RT-PCR in uninjured or injured kidneys. (**D**) *lhx1a* mRNA levels were evaluated by RT-PCR in *wnt4a⁻/⁻* and WT zebrafish kidneys at 7 dpi. *β-actin* was used as a sample control. (**E–H**) *lhx1a* whole-mount in situ hybridization (WISH) showing the trunk kidney region at 7 dpi. (**G**) the number of *lhx1a⁺* cell aggregates in *wnt4a⁻/⁻* was less than that in WT. (**H**) Injection of dmPGE2 could rescue the influence of Wnt4a deficiency. (**I**) *lhx1a⁺* cell aggregates of whole kidney were calculated using ImageJ. n = 5–7 fish for each condition. Data were analyzed by ANOVA, ***p<0.001; ns, no significant difference. (**J, K**) Gentamicin induced *lhx1a⁺* new nephrons with proliferating EdU⁺ nuclei at 5 dpi (n = 5). (**L, M**) The proliferation of *lhx1a⁺* cells in *wnt4a⁻/⁻* was significantly less than that in WT (n = 6). (**N, O**) dmPGE2 could rescue the effect of Wnt4a deficiency and recover the proliferation of *lhx1a⁺* cells in *wnt4a⁻/⁻* (n = 6). (**K, M, O**) show the higher-magnification images of the boxed areas showed in (**J, L, N**). Scale bar in (**J–O**), 100 µm. (**G**) Proliferation ratio of *lhx1a⁺* RPCs in (**J–O**) was calculated using ImageJ. n = 5–7 in each condition. Data were analyzed by ANOVA, ***p<0.001; ns, no significant difference.

*Figure 5 continued on next page*

*Figure 5 continued*

The online version of this article includes the following source data and figure supplement(s) for figure 5:

**Source data 1.** Original gel files of *Figure 5B–D*.

**Source data 2.** Numerical data for *Figure 5A, I, and P and Figure 5—figure supplement 1A*.

**Figure supplement 1.** PGE2 affects Wnt activity during nephron regeneration.

detect the change of the Wnt/β-catenin target gene *lef1*. FISH analyses showed that the reduced Wnt/β-catenin signaling in *cox2a*[-/-]and *wnt4a*[-/-] mutants caused a decrease in *lef1* expression in RPC aggregates, and dmPGE2 treatment rescued the reduced *lef1* expression (*Figure 6—figure supplement 3A–I*). Altogether, these results indicate that PGE2–EP4 signaling regulates Wnt activity in the *lhx1a*[+] RPC aggregate by regulating nuclear β-catenin levels during nephron regeneration.

It has been shown that PGE2 can stabilize β-catenin through the cAMP-PKA (cyclic-AMP-dependent protein kinase A) pathway during hematopoietic stem cell self-renewal, bone marrow repopulation, and liver regeneration in zebrafish and mice (*Goessling et al., 2009*; *Evans, 2009*; *North et al., 2010*; *Brudvik et al., 2011*; *Wang et al., 2016*). To test whether this mechanism is conserved in nephron regeneration, we blocked PKA by injection of the peptide PKA inhibitor PKI (6–22) amide (PKI) or pharmacological inhibitor H89 and found that both inhibitors decreased β-catenin levels in the RPC aggregates at 5 dpi (*Figure 6U and W*, *Figure 6—figure supplement 1F and H*). Moreover, this reduction could not be rescued by the dmPGE2 injection (*Figure 6V and W*, *Figure 6—figure supplement 1G and H*). Furthermore, PKI or H89 also significantly restrained nephron regeneration and dmPGE2 failed to rescue these regeneration defects (*Figure 7A–I*). It has been shown that PKA regulates Wnt activity via direct changes in the phosphorylation status of β-catenin at Ser675 (p-S675-β-catenin) or GSK3β at Ser9 (p-S9-GSK3β) (*Wang et al., 2016*; *Parida et al., 2016*; *Lin et al., 2021*). The S675 phosphorylation of β-catenin stabilizes the β-catenin protein by inhibiting destruction, while p-S9-GSK3β can prevent the assembly of the β-catenin destruction complex (*Hino et al., 2005*; *Taurin et al., 2006*; *Fang et al., 2000*). Therefore, we used the corresponding antibodies to detect changes of these two protein phosphorylation levels in *lhx1a*[+] cell aggregates during RPC aggregation or proliferation. The levels of p-S9-GSK3β or p-S675-β-catenin in *cox2a*[-/-] mutants and PKA inhibitor treatment groups were less than DMSO-treated control groups following AKI, respectively (*Figure 7J–L, Q–S, X, Y*). dmPGE2 treatment rescued the reduction of both levels of p-S9-GSK3β and p-S675-β-catenin in injured *cox2a*[-/-] kidneys, but not in PKA inhibitor-treated groups (*Figure 7M–P, T–W, X, Y*). Collectively, these data indicate that PKA functions downstream of PGE2–EP4 signaling to regulate the stability of β-catenin and nephron regeneration.

## Discussion

Our studies have unveiled that RICs form a cellular network that wraps the RPC aggregate during nephron regeneration, in which RICs promote the proliferation of nephrons by secreting PGE2 (*Figure 8A*). Mechanistically, PGE2 binds to EP4b receptor, which is on the surface of RPCs, resulting in the activation of PKA. PKA phosphorylates β-catenin at Ser675 and GSK3β at Ser9, thereby increasing the β-catenin level in RPCs. Interestingly, Wnt4a that secreted by RPCs can also reduce the stability of the β-catenin destruction complex, thereby enhancing stabilization of β-catenin. (*Figure 8B*). Thus, PGE2 can cooperate with Wnt4a to regulate the level of β-catenin in RPCs. The increase in β-catenin is necessary for the proliferation of RPCs during nephron regeneration. Hence, RICs provide a microenvironment for the proliferation of RPCs, which is necessary for the rapid recovery of damaged nephrons.

RICs can be labeled with *Tg(FoxD1-Cre; R26R)* and *Tg(Tenascin-C-CreER2; ROSA26-lacZ)* transgenic lines in mice (*Humphreys et al., 2010*; *He et al., 2013*). However, the *FoxD1* expression was hardly detectable in zebrafish kidney single-cell sequencing data (*Figure 1—source data 1*). Therefore, new markers need to be identified and developed for zebrafish RICs. Our single-cell sequencing analyses of the zebrafish kidney identified *fabp10a* as a RIC marker. Subsequently, we proved that RICs were specifically labeled by the *fabp10a* promoter-driven GFP in the *Tg(fabp10a:GFP)* transgenic line, a major step toward studies of the development, function, and regeneration of zebrafish kidney. Although *fabp10a* is expressed only in zebrafish interstitial cells, *fabp3*, *fabp4*, and *fabp5* are

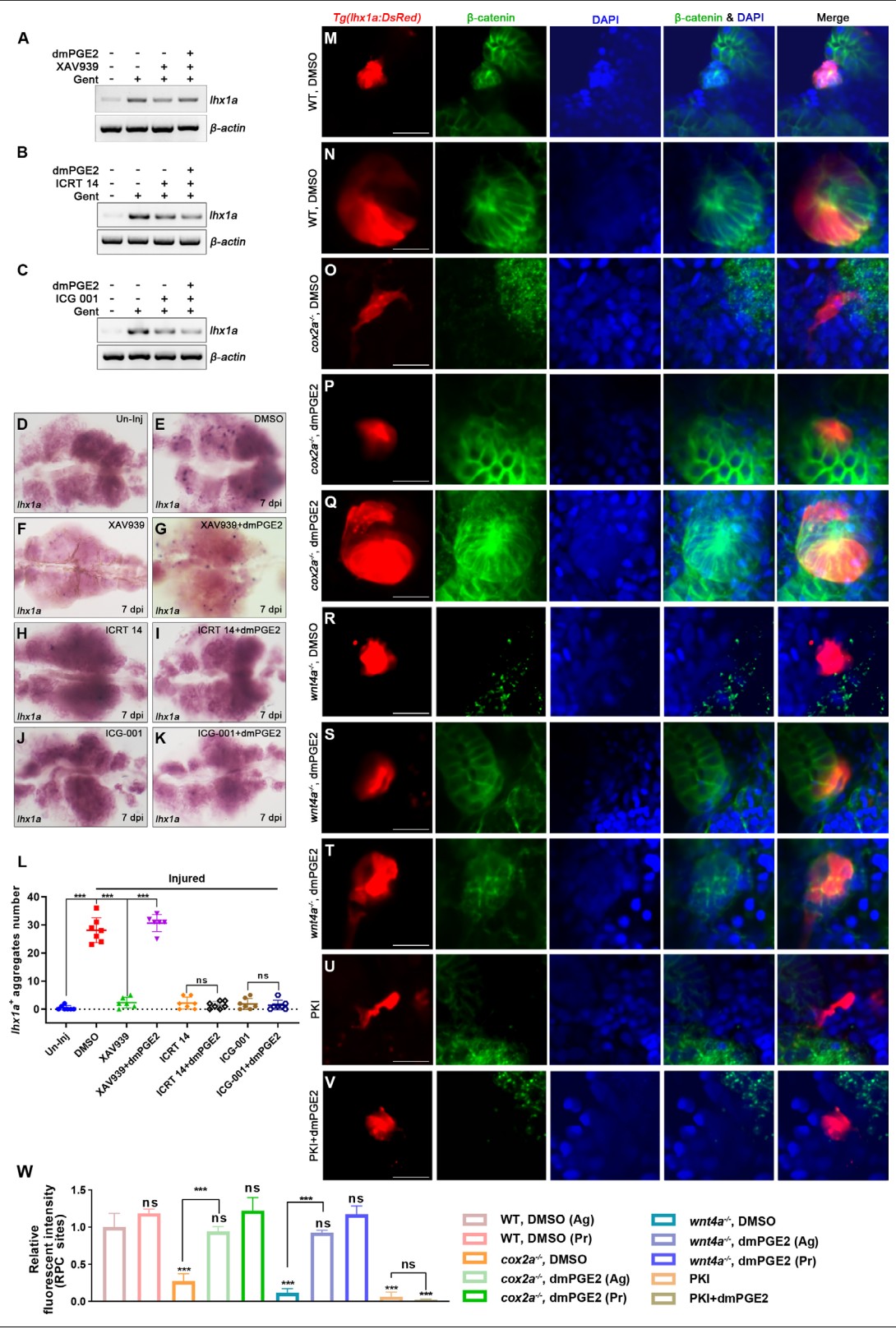

**Figure 6.** PGE2–EP4 signaling promotes nephron regeneration through regulating β-catenin level. (**A–C**) *lhx1a* mRNA levels were assessed by RT-PCR at 7 dpi. *β-actin* was used as a sample control. (**D–K**) *lhx1a* whole-mount in situ hybridization (WISH) showing the trunk kidney region at 7 dpi. XAV939 (**F**) could reduce the number of *lhx1a* $^+$ cell aggregates and dmPGE2 could rescue the influence of XAV939 treatment (**G**); ICRT 14 (**H**) or ICG-001 (**J**) could reduce the number of *lhx1a* $^+$ cell aggregates and dmPGE2 could not rescue the influence of ICRT 14 (**I**) or ICG-001 (**K**) treatment. n = 5–7 in

*Figure 6 continued on next page*

*Figure 6 continued*

each condition. (**L**) *lhx1a*+ cell aggregates of whole kidney were calculated using ImageJ. Data were analyzed by ANOVA, ***p<0.001; ns, no significant difference. (**M–V**) Immunofluorescence staining of β-catenin in *Tg(lhx1a:DsRed)* zebrafish kidneys at 5 dpi. (**M, N**) Zebrafish injected DMSO as a control group, and the amount of β-catenin could be detected in *lhx1a*+ cell aggregates during renal progenitor cell (RPC) aggregation (**M**) or proliferation (**N**). (**O**) β-catenin level in *lhx1a*+ cell aggregates of *cox2a*-/- was significantly less than the control group, and injection of dmPGE2 (**P, Q**) could rescue the influence of Cox2a deficiency. (**R**) β-catenin level in *lhx1a*+ cell aggregates of *wnt4a*-/- was significantly less than the control group, and injection of dmPGE2 (**S, T**) could rescue the influence of Wnt4a deficiency. Injection of PKI (**U**) could reduce β-catenin level in *lhx1a*+ cell aggregates, while injection of dmPGE2 (**V**) could not rescue the influence of PKI treatment. n = 3–6 in (**M–V**). Scale bar in (**M–V**), 50 μm. (**W**) Bar chart depicting β-catenin levels following acute kidney injury (AKI) (**M–V**). Fluorescent intensities per unit area were measured at the *lhx1a*+ RPC aggregates using ImageJ. β-catenin levels of *lhx1a*+ RPCs during RPC aggregation normalized as 1. Ag, aggregation; Pr, proliferation. n = 3–6 in each condition. Data were analyzed by ANOVA, ***p<0.001; ns, no significant difference.

The online version of this article includes the following source data and figure supplement(s) for figure 6:

**Source data 1.** Original gel files of *Figure 6A–C*.

**Source data 2.** Numerical data for *Figure 6L and W*, *Figure 6—figure supplement 1H*, *Figure 6—figure supplement 2F*, and *Figure 6—figure supplement 3I*.

**Figure supplement 1.** PGE2–EP4 signaling promotes nephron regeneration through regulating β-catenin level.

**Figure supplement 2.** PGE2-EP4 signal regulates β-catenin levels in renal progenitor cells (RPCs).

**Figure supplement 3.** PGE2–EP4 signaling promotes nephron regeneration through regulating Wnt signaling activity.

highly expressed in mouse and human RICs (*Young et al., 2018*; *Stewart et al., 2019*; *Ransick et al., 2019*), suggesting a concretive role of fatty acid-binding protein family members in marking RICs. During nephron regeneration, many RICs were EdU+, which proved that RICs mainly came from self-proliferation. Whether there is a RIC progenitor cell population in adult zebrafish kidney needs to be assessed in future studies.

Taking advantage of real-time observations in zebrafish juveniles, we find the interactions between RICs and RPCs and discovered that RICs form a network wrapping the entire RPC aggregates. Notably, RICs highly express Cox2a and secrete PGE2 during regeneration, and most of the RPC aggregates fail to proliferate and differentiate normally when PGE2 production is inhibited. These results demonstrate that the RIC network provides a necessary microenvironment for the RPCs, in which PGE2 is a crucial signaling molecule to rapidly induce the formation of new nephrons. We find that the PGE2 receptor EP4b is highly expressed in RPCs, and PGE2 acts through the EP4 receptor during nephron regeneration. Importantly, we demonstrate that PGE2–EP4 signaling interacts with Wnt signaling to promote nephron regeneration. Wnt4 is a key regulator of the MET during nephrogenesis in the mouse kidney (*Carroll et al., 2005*; *Stark et al., 1994*; *Kispert et al., 1998*). In zebrafish, the *wnt4a*-/- mutants exhibit a nephron regeneration defect, and the RPC aggregates fail to proliferate and differentiate into RVs. The dmPGE2 injection can rescue the regeneration defect in Wnt4a-deficient kidney by stabilizing β-catenin, revealing the cooperation of multiple signaling pathways during nephron regeneration.

Overall, our study enriches the understanding of the PGE2/Wnt signaling interaction and renal cell collaboration during nephrogenesis and nephron regeneration, which provides unique insights into developing regenerative strategies for potential interventions in AKI and renal diseases.

# Materials and methods

**Key resources table**

| Reagent type (species) or resource | Designation | Source or reference | Identifiers | Additional information |
|---|---|---|---|---|
| Gene (*Danio rerio*) | *cox2a* | GenBank | NM_153657.1 | |
| Gene (*D. rerio*) | *cox1* | GenBank | NM_153656.2 | |
| Gene (*D. rerio*) | *cox2b* | GenBank | NM_001025504.2 | |
| Gene (*D. rerio*) | *ep1* | GenBank | NM_001166330.1 | |
| Gene (*D. rerio*) | *ep2a* | GenBank | NM_200635.1 | |

*Continued on next page*

*Continued*

| Reagent type (species) or resource | Designation | Source or reference | Identifiers | Additional information |
|---|---|---|---|---|
| Gene (*D. rerio*) | *ep3* | GenBank | XM_017356646.2 | |
| Gene (*D. rerio*) | *ep4a* | GenBank | NM_001039629.1 | |
| Gene (*D. rerio*) | *ep4b* | GenBank | NM_001128367.1 | |
| Gene (*D. rerio*) | *wnt4a* | GenBank | NM_001040387.1 | |
| Gene (*D. rerio*) | *lhx1a* | GenBank | NM_131216.1 | |
| Gene (*D. rerio*) | *lef1* | GenBank | NM_131426.1 | |
| Gene (*D. rerio*) | *epoa* | GenBank | NM_001038009.2 | |
| Strain, strain background (*D. rerio*) | AB | Laboratory resources | Labs | |
| Strain, strain background (*D. rerio*) | Tg(fabp10a:GFP) | Laboratory resources | Labs | |
| Strain, strain background (*D. rerio*) | Tg(lhx1a:DsRed) | This study | | methods |
| Strain, strain background (*D. rerio*) | Tg(cdh17:DsRed) | This study | | methods |
| Strain, strain background (*D. rerio*) | wnt4a$^{-/-}$ | Chinese National Zebrafish Resource Center (Wuhan, China) | fh294 | |
| Strain, strain background (*D. rerio*) | cox2a$^{-/-}$ | *Li et al., 2019* | | |
| Antibody | Anti-Cox2 (goat polyclonal) | Cayman, 100034-lea | RRID:AB_10078977 | IF (1:200) |
| Antibody | Anti-β-catenin (mouse monoclonal) | Sigma, C7207 | RRID:AB_2086128 | IF (1:200) |
| Antibody | Anti-p-Ser9-GSK3 beta (rabbit polyclonal) | Abcam, ab107166 | RRID:AB_476865 | IF (1:200) |
| Antibody | Anti-phospho-β-catenin (ser675) (D2F1) XP (rabbit monoclonal) | CST, 4176T | RRID:AB_1903923 | IF (1:200) |
| Antibody | Anti-Pax2a (rabbit polyclonal) | Abcam, ab229318, *Chen et al., 2019* | | IF (1:200) |
| Antibody | Anti-Pan-cadherin (rabbit polyclonal) | Sigma, C3678 | RRID:AB_258851 | IF (1:100) |
| Antibody | Goat anti-mouse IgG H&L Alexa Fluor 647 (goat polyclonal) | Abcam, ab150115 | RRID:AB_2687948 | IF (1:500) |
| Antibody | Donkey anti-goat IgG (Alexa Fluor 647) (donkey polyclonal) | Abcam, ab150131 | RRID:AB_2732857 | IF (1:500) |
| Antibody | Goat anti-rabbit IgG (H+L) Alexa Fluor 488 (goat polyclonal) | Invitrogen, A11008 | RRID:AB_143165 | IF (1:500) |
| Antibody | Goat anti-rabbit IgG (H+L) Alexa Fluor 633 (goat polyclonal) | Invitrogen, A21070 | RRID:AB_2535731 | IF (1:500) |
| Chemical compound, drug | Indomethacin | Sigma | I7378-5G | |
| Chemical compound, drug | NS-398 | Sigma | N194-5MG | |
| Chemical compound, drug | TG4-155 | Selleck | S6793 | |
| Chemical compound, drug | dmPGE2 | Sigma | D0160 | |
| Chemical compound, drug | GW627368X | TOPSCIENCE | T1978 | |
| Chemical compound, drug | XAV939 | Selleck | S1180 | |
| Chemical compound, drug | ICRT 14 | MCE | HY-16665 | |

*Continued on next page*

*Continued*

| Reagent type (species) or resource | Designation | Source or reference | Identifiers | Additional information |
|---|---|---|---|---|
| Chemical compound, drug | ICG-001 | MCE | HY-14428 | |
| Chemical compound, drug | H89 (dihydrochloride) | MCE | HY-15979A | |
| Chemical compound, drug | PKA Inhibitor Fragment (6-22) amide (TFA) | MCE | HY-P1290A | |
| Software, algorithm | GraphPad Prism (version 8.02) | GraphPad Prism, version 8.02 | RRID:SCR_002798 | |
| Software, algorithm | Excel 2019 | Microsoft, version office home and student 2019 | RRID:SCR_016137 | |
| Software, algorithm | ImageJ for Windows, V 1.8.0 | National Institutes of Health | RRID:SCR_001935 | |
| Other | DAPI Stain Solution | Beyotime | C1002 | For nucleic acid staining |

## Zebrafish

Zebrafish embryos, larvae, and adults were produced, grown, and maintained according to standard protocols described in the zebrafish book. For experiments with adult zebrafish, animals ranging in age from 3 to 12 months were used. Zebrafish were maintained in standard conditions under a 14-hr-light and 10-hr-dark cycle and fed twice daily. Zebrafish were anesthetized using 0.0168% buffered tricaine (MS-222, Sigma). AB strain of zebrafish is used as the wild-type (WT). The following zebrafish lines were used in this study: *Tg(fabp10a:GFP)* (*Her et al., 2003*); *cox2a^-/-* (*Li et al., 2019*); *wnt4a^-/-* (fh294) was purchased from the Chinese National Zebrafish Resource Center (Wuhan, China); and *Tg(cdh17:DsRed)* and *Tg(lhx1a:DsRed)* were constructed in this study.

## Generation of *Tg(lhx1a:DsRed)* transgenic line

We obtained the *lhx1a:EGFP*/pI-SceI plasmid from Dr. Neil A Hukdiede (*Swanhart et al., 2010*), substituted DsRed for EGFP using ClonExpress II One Step Cloning Kit (Vazyme, C112-01), and constructed *lhx1a:DsRed*/pI-SceI plasmid. Then, 30 pg of *lhx1a:DsRed*/pI-SceI plasmid DNA was injected into one-cell-stage embryos along with the I-SceI restriction enzyme (NEB, R0694S). These injected embryos were raised to adulthood and screened for DsRed expression in known *lhx1a* expression domains, and a stable *Tg(lhx1a:DsRed)* transgenic line was isolated.

## Generation of *Tg(cdh17:DsRed)* transgenic line

Promoters of *cadherin-17* (*cdh17*) (–4.3k) were amplified from zebrafish genomic DNA by PCR and constructed *cdh17:DsRed*/pI-SceI plasmids (*Liao et al., 2021*). These injected embryos were raised to adulthood and screened for DsRed expression in known *cdh17* expression domains, and isolated stable *Tg(cdh17:DsRed)* transgenic line was constructed.

## Zebrafish AKI model

Intraperitoneal injection of gentamicin was used to induce AKI as previously described (*Chen et al., 2019*). In brief, gentamicin (2.7 µg/µL, 20 µL per fish) diluted in water was intraperitoneally injected in WT or other zebrafish lines. Each injected zebrafish was then dropped into an individual container. The fish excreting proteinuria at 1 dpi was used for subsequent experiments.

## Single-cell RNA sequencing

Kidney cells from six randomly selected 6-month-old zebrafish were loaded into Chromium microfluidic chips with 30v chemistry and barcoded with a 10× Chromium Controller (10X Genomics). RNA from the barcoded cells was subsequently reverse-transcribed and sequencing libraries were constructed with reagents from a Chromium Single Cell 30v3 reagent kit (10X Genomics) according to the manufacturer's instructions. Sequencing was performed on an Illumina (NovaSeq) platform according to the manufacturer's instructions (Illumina). The Seurat package was used for data normalization, dimensionality reduction, clustering, and differential expression. We used Seurat alignment method canonical correlation analysis (*Butler et al., 2018*) for integrated analysis of datasets. For clustering, highly

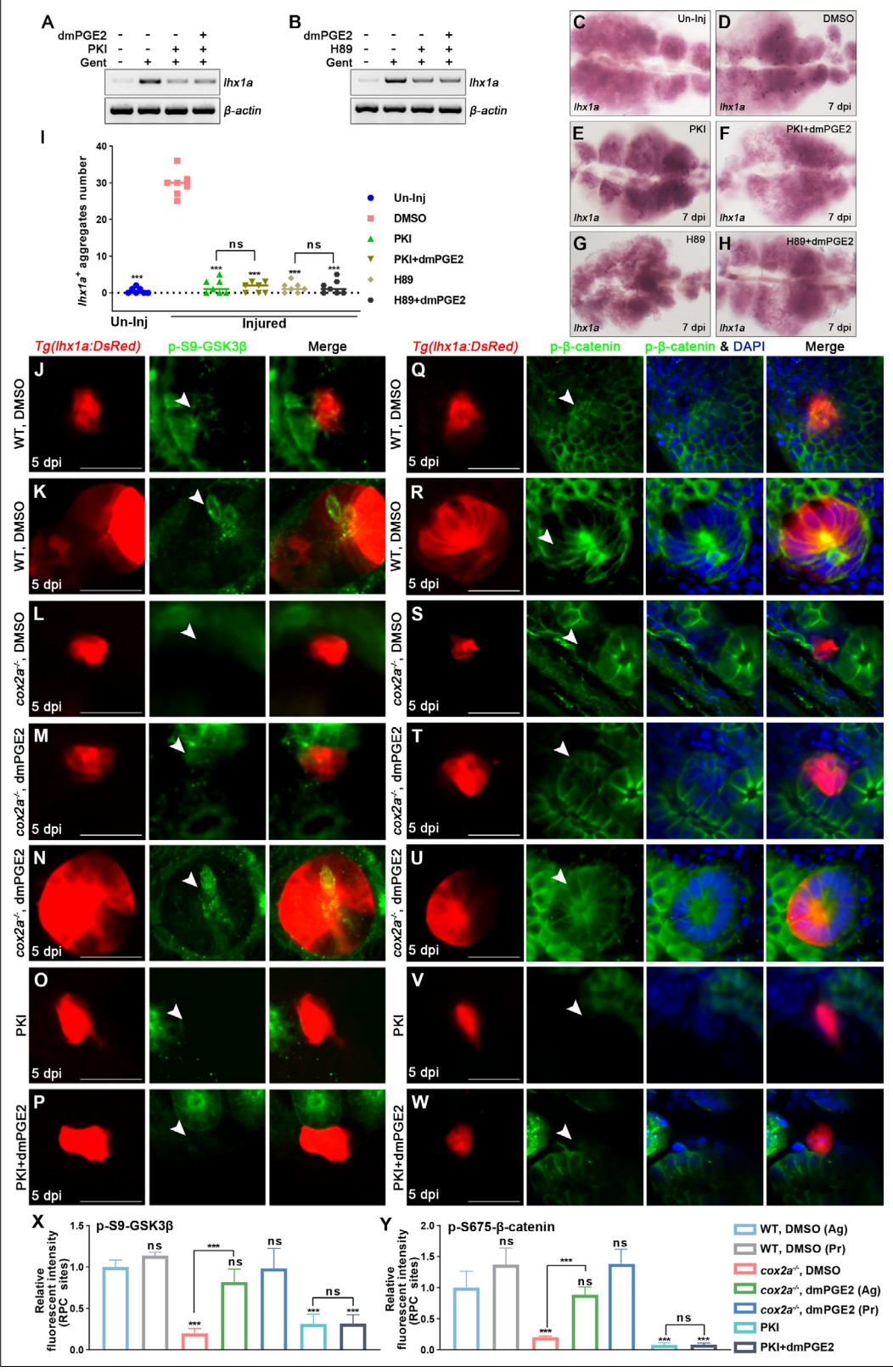

**Figure 7.** PGE2 regulates the stability of β-catenin in renal progenitor cells (RPCs) through PKA. (**A, B**) *lhx1a* mRNA levels were evaluated by RT-PCR at 7 dpi. *β-actin* was used as a sample control. (**C–H**) *lhx1a* whole-mount in situ hybridization (WISH) showing the trunk kidney region at 7 dpi. PKI (**E**) or H89 (**G**) treatment reduced the number of *lhx1a*+ cell aggregates, while injection of dmPGE2 could not rescue the influence of PKI (**F**) or H89 (**H**) treatment.

*Figure 7 continued on next page*

*Figure 7 continued*

(**I**) *lhx1a+* cell aggregates of whole kidney were calculated using ImageJ. n = 5–7 in each condition. Data were analyzed by ANOVA, ***p<0.001; ns, no significant difference. (**J–N**) Immunofluorescence staining of p-S9-GSK3β (arrowheads) in *Tg(lhx1a:DsRed)* zebrafish kidneys at 5 dpi. (**J, K**) Zebrafish injected with DMSO as a control group, and the amount of p-S9-GSK3β could be detected in *lhx1a+* cell aggregates cytoplasm during RPC aggregation (**J**) or proliferation (**K**). (**L**) p-S9-GSK3β in *lhx1a+* cell aggregates of *cox2a-/-* was hardly detectable, and injection of dmPGE2 (**M, N**) could rescue the influence of Cox2a deficiency. Injection of PKI (**O**) could reduce p-S9-GSK3β level in *lhx1a+* cell aggregates, while injection of dmPGE2 (**P**) could not rescue the influence of PKI treatment. (**Q–W**) Immunofluorescence staining of p-S675-β-catenin in *Tg(lhx1a:DsRed)* zebrafish kidneys at 5 dpi. (**Q, R**) Injection of DMSO as a control group and amounts of p-S675-β-catenin could be detected in *lhx1a+* cell aggregates during RPC aggregation (**Q**) or proliferation (**R**). (**S**) p-S675-β-catenin level in *lhx1a+* cell aggregates of *cox2a-/-* was hardly detectable, and injection of dmPGE2 (**T, U**) could rescue the influence of Cox2a deficiency. Injection of PKI (**V**) could reduce p-S675-β-catenin level in *lhx1a+* cell aggregates, while injection of dmPGE2 (**W**) could not rescue the influence of PKI treatment. Scale bar, 50 µm. (**X, Y**) Bar chart depicting p-S9-GSK3β (**X**) and p-S675-β-catenin (**Y**) levels following acute kidney injury (AKI) (**J–W**). Fluorescent intensities per unit area were measured at the *lhx1a+* RPC aggregates using ImageJ. p-S9-GSK3β or p-S675-β-catenin levels of *lhx1a+* RPCs during RPC aggregation normalized as 1. Ag, aggregation; Pr, proliferation. n = 3–6 in each condition. Data were analyzed by ANOVA, ***p<0.001; ns, no significant difference.

The online version of this article includes the following source data for figure 7:

**Source data 1.** Original gel files of *Figure 7A and B*.

**Source data 2.** Numerical data for *Figure 7I, X, and Y*.

variable genes were selected and the principal components based on those genes were used to build a graph, which was segmented with a resolution of 0.6. GO enrichment analysis of marker genes was implemented using the cluster Profiler R package, in which gene length bias was corrected. GO terms with corrected p-values <0.05 were considered significantly enriched by the marker gene of interest. Sequencing data have been deposited in GEO under accession code GSE183382.

## Transcriptome sequencing

Six-month-old zebrafish were used for RNA sequencing. Kidney samples were obtained at 0, 1, 3, 5, and 7 dpi and RNA was isolated from kidney tissue using TRIzol reagent (Invitrogen, 15596018) according to the manufacturer's protocol. A total amount of 1 µg RNA per sample was used as input material for the RNA sample preparations. Sequencing libraries were generated using NEB Next Ultra RNA Library Prep Kit for Illumina (NEB, USA) following the manufacturer's recommendations and index codes were added to attribute sequences to each sample. The clustering of the index-coded samples was performed on a cBot Cluster Generation System using TruSeq PE Cluster Kit v3-cBot-HS (Illumia) according to the manufacturer's instructions. After cluster generation, the library preparations were sequenced on an Illumina NovaSeq platform and 150 bp paired-end reads were generated. Raw data (raw reads) of fastq format were firstly processed through in-house Perl scripts. In this step, clean data (clean reads) were obtained by removing reads containing adapter, reads containing ploy-N and low-quality reads from raw data. At the same time, Q20, Q30, and GC content, the clean data were calculated. All the downstream analyses were based on the clean data with high quality. The mapped reads of each sample were assembled by StringTie (v1.3.3b) (*Pertea et al., 2015*) in a reference-based approach. FeatureCounts v1.5.0-p3 was used to count the reads numbers mapped to each gene. And then FPKM (expected number of fragments per kilobase of transcript sequence per million base pairs sequenced) of each gene was calculated based on the length of the gene and reads count mapped to this gene. Differential expression analysis of two conditions/groups (three biological replicates per condition) was performed using the DESeq2 R package (1.16.1). The resulting p-values were adjusted using the Benjamini and Hochberg's approach for controlling the false discovery rate. Genes with an adjusted p-value<0.05 found by DESeq2 were assigned as differentially expressed. Sequencing data have been deposited in GEO under accession code GSE191068.

## ELISA for PGE2

ELISA for PGE2 was conducted using a Prostaglandin E2 ELISA Kit (D751014-0048, BBI) according to the manufacturer's instructions. Briefly, kidneys were collected from 20 weight-matched fish, washed

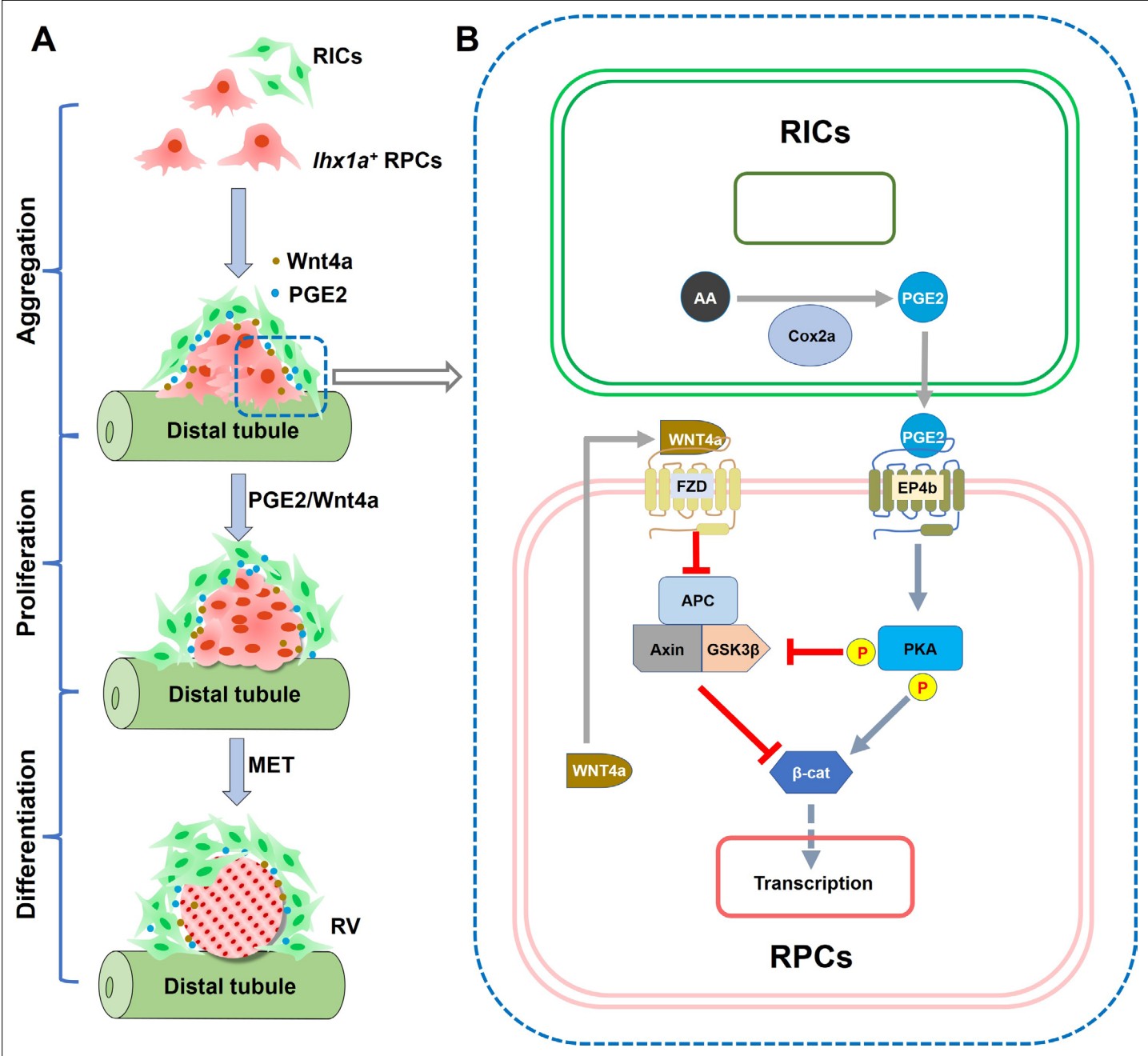

**Figure 8.** Graphical abstract summarizing the research findings. (**A**) Diagram of RV formation process. During nephron regeneration, renal progenitor cells (RPCs) congregate to form cell aggregate, then renal interstitial cells (RICs) form a network to wrap the RPC aggregate and secrete PGE2. RIC-secreted PGE2 and RPC-secreted Wnt4a synergistically promote RPCs to proliferate rapidly and then differentiate into RV. (**B**) Model of the interaction between PGE2 and Wnt signaling pathway in RPC during regeneration. During nephron regeneration, RICs that are in close contact with RPC aggregates express Cox2a and secrete PGE2. PGE2 signaling interacts with Wnt/β-catenin pathway at the level of β-catenin destruction complex and direct modification of β-catenin stability in RPC. This is achieved through EP4b induction and the activation of PKA. PKA phosphorylates GSK3β at Ser9, preventing the assembly of the β-catenin destruction complex. PKA also phosphorylates β-catenin at Ser675, resulting in stabilization of β-catenin. Wnt4a secreted by RPC can also reduce the stability of the destruction complex through the FZD receptor, thereby enhancing stabilization of β-catenin in response to acute kidney injury (AKI).

in ice-cold phosphate-buffered saline (PBS), and homogenized in 500 µL PBS. Homogenate was spun down at 12,000 rpm for 10 min at 4°C to eliminate particulate, and the supernatant was collected for ELISA. Assays were run in technical triplicate.

## Inhibitor treatment

During the nephron regeneration stage, the Indo (I7378-5G, Sigma; 400 µM, 10 µL per fish), NS-398 (N194-5MG, Sigma; 140 µM, 10 µL per fish), TG4-155 (S6793, Selleck; 400 µM, 10 µL per fish), dmPGE2 (D0160, Sigma; 600 µM, 10 µL per fish), GW627368X (T1978, TOPSCIENCE; 200 µM, 10 µL per fish), XAV939 (S1180, Selleck; 200 µM, 10 µL per fish), ICRT 14 (HY-16665, MCE; 10 µM, 10 µL per fish), ICG-001 (HY-14428, MCE; 100 µM, 10 µL per fish), PKI (HY-P1290A, MCE; 20 µM, 10 µL per fish), and H89 (HY-15979, MCE; 200 µM, 10 µL per fish) were intraperitoneally injected into zebrafish at 2, 4, and 6 dpi. Kidneys were collected at 7 dpi for RNA extraction, WISH, or immunofluorescence. For the 5 dpi test, inhibitors were injected at 2 and 4 dpi, and samples were collected at 5 dpi; 1% DMSO was injected in control groups with the same conditions. For juvenile zebrafish experiments, dmPGE2 (2 µM in egg water) was used to test the influence on mesonephros development. 0.2% DMSO was used in control groups under the same conditions.

## WISH

WISH was performed as previously described using *lhx1a* and *ep4b* probes (*Chen et al., 2019*). Briefly, fish with internal organs and hands removed were fixed overnight in 4% paraformaldehyde. Fixed kidneys were removed from body and permeabilized with proteinase K (10 µg/mL, Roche) in PBT (0.1% tween-20 in PBS) for 1 hr with rocking. Digoxigenin-labeled riboprobes were generated from cDNA fragments comprising the sequences of zebrafish *lhx1a* or *ep4b probe*. Hybridization was performed as previously described (*Chen et al., 2019*). Anti-DIG AP antibody and NBT/BCIP substrate (Roche) were used to detect the probe. After the color reaction, images were taken using a BX3-CBH microscope (Olympus, Japan).

## Fluorescence in situ hybridization

Fluorescence in situ hybridization was performed as previously described using *lef1* probes (*He et al., 2020*). Briefly, fish with internal organs and hands removed were fixed overnight in 4% paraformaldehyde. kidneys were harvested and embedded in OCT to obtain frozen-sections at 100 µm. Sections permeabilized with proteinase K (10 µg/mL, Roche) in PBT for 20 min with rocking. Digoxigenin-labeled riboprobes were generated from cDNA fragments comprising the sequences of zebrafish *lef1 probe* (F: 5'-ATGCCGCAGTTGTCAGGTG-3', R: 5'-CGCTTTCCTCCATTGTTCAGATG-3'). Hybridization was performed as previously described (*He et al., 2020*). Anti-DIG POD antibody (11207733910, Roche) and TSA plus fluorescein system (NEL741001KT, PerkinElmer) were used to detect the probe. Fluorescent intensities per unit area were measured using ImageJ.

## EdU assay

A Click-iT Plus EDU Alexa Fluor 647 Imaging Kit (C10640, Invitrogen) was used to detect cell proliferation in juvenile zebrafish or sections of kidney. Briefly, EdU solution (200 mM, 10 µL per fish) was intraperitoneally injected into fish. After 3 hr, kidneys were obtained and proliferation measurements were performed. Quantification of EdU was performed in a blinded manner using ImageJ, Briefly, 4-square mm kidney images were taken randomly. Subsequently, Total number of *lhx1a*[+] cells and *lhx1a*[+] EdU[+] cells were calculated for further study. Images were marked so that the person performing the analysis was unaware of the treatment conditions for each sample prior to calculation.

## Semi-quantitative RT-PCR and quantitative PCR

RNA was isolated from kidney tissue using TRIzol reagent (15596018, Invitrogen) according to the manufacturer's protocol. A Prime Script II first-strand cDNA Synthesis Kit (9767, Takara) was used to synthesize cDNA, which was then subjected to PCR using Taq Master Mix (p112-01, Vazyme) for RT-PCR or TB Green Premix EX Taq II (Takara, RR820A) for quantitative PCR. The following primers were used: *lhx1a* (F: gacaggtttctccttaatgttc, R: CTTTCAGTGTCTCCAGTTGC); *cox2a*, (F: CGCTATAT CCTGTTGTCAAGG, R: gatggtctcaccaatcagg); *col1a1b*, (F: GGTTCTGCTGGTAACGATGG, R: CCAG GCATTCCAATAAGACC); *col1a2*, (F: CTGGTAAAGATGGTTCAAATGG, R: CACCTCGTAATCCTTG

GCT); *lef1 Kamei et al., 2019*; *cox1* and *cox2b* (*FitzSimons et al., 2020*). The primers for *ep1a* were QF: AAATGTCACCTCGAGCAGAC, QR: ACAGGAGAAAGGCCTTGGAT; those for *ep4b* were QF: ATCGTTCTCATAGCCACGTCCACT, QR: CCGGGTTTGGTCTTGCTGATGAAT, other *eps* primers were as previously described (*FitzSimons et al., 2020*). *β-actin* (*Chen et al., 2019*) or *rpl13a* (*FitzSimons et al., 2020*) were used to standardize samples.

## FACS

To obtain RNA libraries of *lhx1a⁺* cells or RICs, *Tg(lhx1a:DsRed)* or *Tg(fabp10a:GFP;cdh17:DsRed)* kidney cells from 10 fish were manually dissected in 1% PBS and 0.005% trypsin-EDTA solution at 0 or 5 dpi. Cells of interested were sorted using MoFlo XDP (Beckman) and collected for RNA extraction.

## Immunofluorescence

For immunofluorescent analysis, kidneys were harvested and fixed in 4% formaldehyde overnight at 4°C, and then embedded in OCT (optimal cutting temperature compound) to obtain frozen-sections at 100 μm on Micron HM550 cryostat. The primary antibodies used were anti-Cox2 (100034-lea, Cayman), anti-Pax2a (ab229318, Abcam), anti-β-catenin (C7207, Sigma), anti-p-Ser9-GSK3 beta (ab107166, Abcam), anti-Pan-cadherin (C3678, Sigma), and anti-phospho-β-catenin (ser675) (D2F1) XP Rabbit mab (4176T, CST). The secondary antibodies used goat anti-mouse IgG H&L Alexa Fluor 647 (ab150115, Abcam), donkey anti-goat IgG Alexa Fluor 647 (ab150131, Abcam), goat anti-rabbit IgG (H+L) Alexa Fluor 633 (A11008, Invitrogen), and goat anti-rabbit IgG (H+L) Alexa Fluor 488 (A21070, Invitrogen), at 1:500. Images were taken using the Nikon A1 confocal microscope. Fluorescent intensities per unit area were measured using ImageJ.

## Nile red staining

For Nile red staining, *Tg(fabp10a:GFP)* kidneys were harvested and fixed in 4% formaldehyde overnight at 4°C, and then embedded in OCT to obtain frozen-sections at 10 μm on Micron HM550 cryostat. Sections were incubated for 5 min in 1 μM Nile red-PBS buffer at room temperature and then washed three times using PBS. Images were taken using the Nikon A1 confocal microscope.

## Statistics

All data are presented as means ± standard deviation (SD). Unless otherwise specified, all experiments were carried out using more than three independent replicates. Statistical analysis was performed using GraphPad Prism (version 8.02) and Excel (version, Microsoft office home and student 2019) for Microsoft Windows. Data were analyzed by ANOVA, *p<0.05, **p<0.01, ***p<0.001; ns, no significant difference.

# Acknowledgements

We thank Dr. Neil Hukriede for sending us the *lhx1a:EGFP*/pI-SceI plasmid. This work was funded by the National Key Research and Development Program of China (2017YFA0106600, 2018YFA0800103, 2018YFA0801004) and the National Natural Science Foundation of China (32070822, 82030023, 31771609, 31970780).

# Additional information

### Funding

| Funder | Grant reference number | Author |
| --- | --- | --- |
| National Key Research and Development Program of China | 2017YFA0106600 | Chi Liu |
| National Natural Science Foundation of China | 32070822 | Chi Liu |

| Funder | Grant reference number | Author |
|---|---|---|
| National Natural Science Foundation of China | 82030023 | Jinghong Zhao |
| National Natural Science Foundation of China | 31771609 | Chi Liu |
| National Key Research and Development Program of China | 2018YFA0800103 | Tao P Zhong |
| National Key Research and Development Program of China | 2018YFA0801004 | Tao P Zhong |
| National Natural Science Foundation of China | 31970780 | Tao P Zhong |

The funders had no role in study design, data collection and interpretation, or the decision to submit the work for publication.

### Author contributions

Xiaoliang Liu, Resources, Data curation, Software, Validation, Investigation, Visualization, Methodology, Writing - original draft, Writing - review and editing; Ting Yu, Resources, Data curation, Formal analysis, Validation, Investigation, Visualization, Methodology, Writing - original draft, Writing - review and editing; Xiaoqin Tan, Resources, Data curation, Investigation, Methodology; Daqing Jin, Resources, Investigation, Writing - review and editing; Wenmin Yang, Investigation, Methodology, Writing - original draft, Writing - review and editing; Jiangping Zhang, Lu Dai, Zhongwei He, Resources, Investigation, Methodology; Dongliang Li, Yunfeng Zhang, Shuyi Liao, Investigation, Methodology; Jinghong Zhao, Conceptualization, Investigation, Writing - original draft, Writing - review and editing; Tao P Zhong, Conceptualization, Resources, Investigation, Methodology, Writing - original draft, Writing - review and editing; Chi Liu, Conceptualization, Resources, Data curation, Formal analysis, Supervision, Funding acquisition, Investigation, Methodology, Writing - original draft, Project administration, Writing - review and editing

### Author ORCIDs

Xiaoliang Liu http://orcid.org/0000-0002-4239-9879
Shuyi Liao http://orcid.org/0000-0003-3137-4403
Jinghong Zhao http://orcid.org/0000-0001-9750-3285
Chi Liu http://orcid.org/0000-0003-2057-9649

### Ethics

In this study, all animal care and use protocol was approved by the Institutional Animal Care and Use Committee of the Army Medical University, China (SYXK-PLA-2007035).

### Decision letter and Author response

Decision letter https://doi.org/10.7554/eLife.81438.sa1
Author response https://doi.org/10.7554/eLife.81438.sa2

## Additional files

### Supplementary files

• Transparent reporting form

### Data availability

The authors declare that all data supporting the findings of this study are available within the article and its supplementary information files. Sequencing data have been deposited in GEO under accession codes GSE183382 and GSE191068.

The following datasets were generated:

| Author(s) | Year | Dataset title | Dataset URL | Database and Identifier |
|---|---|---|---|---|
| Liu C, Liu X | 2021 | adult zebrafish kidney cells | https://www.ncbi.nlm.nih.gov/geo/query/acc.cgi?acc=GSE183382 | NCBI Gene Expression Omnibus, GSE183382 |
| Liu C, Liu X | 2021 | Zebrafish kidney regeneration transcriptome | https://www.ncbi.nlm.nih.gov/geo/query/acc.cgi?acc=GSE191068 | NCBI Gene Expression Omnibus, GSE191068 |

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
