## [Editor Report]

This fundamental work substantially advances our understanding of the kidney interstitium and how it influences kidney development focusing on zebrafish as a model organism. The evidence supporting the conclusions is compelling, using single-cell analysis combined with in vivo zebrafish studies to mechanistically explore the functional importance of the discovery. The work will be of broad interest to cell and developmental biologists as well as the kidney community.

---

## [Decision Letter]

**Decision letter after peer review:**

Thank you for submitting your article "Renal Interstitial Cells Promote Nephron Regeneration by Secreting Prostaglandin E2" for consideration by *eLife*. Your article has been reviewed by 2 peer reviewers, and the evaluation has been overseen by a Reviewing Editor and Didier Stainier as the Senior Editor. The following individual involved in the review of your submission has agreed to reveal their identity: Rebecca A. Wingert (Reviewer #1).

Essential revisions:

Please address all the comments of the reviewers. A particular focus should be on the following points:

1) In line with Reviewer #1/point 3 the number of samples analyzed should be indicated and the immunofluorescence should be quantified. In addition, please make sure that all the data are analyzed for statistical analysis when appropriate.

2) In line with the comments of Reviewer #2/point 1 on the single-cell data, the clustering results need to be put into the context with the single-cell data from mouse and if possible from humans.

3) Reviewer #2/point 3 raises some concerns in respect to on the connection between PGE2 signaling and Wnt signaling. Please further clarify the interaction based on the suggestions of the review so that the general readership can appreciate its significance.

*Reviewer #1 (Recommendations for the authors):*

The manuscript is clearly written and was a joy to read. I have not read such a wonderful study in a very long time. I have a few questions and suggestions to improve the work but endorse it very strongly and enthusiastically for publication.

1. Which part of the embryonic pronephros exhibits fabp10a expression at the 4-6 dpf stages (Figure 2, figure supplement 1A)? The relative position (proximal, distal, throughout the tubule) would be helpful to describe.

2. The manuscript would be well-served to cite and acknowledge the formative work of Renate Reimschuessel, who documented neonephrogenesis in fishes and established gentamicin for such studies, in addition to the team of Diep et al. 2011 that adopted this for zebrafish (line 148).

3. In a number of figures, the authors show confocal imaging of representative kidneys stained by immunofluorescence. Two comments related to these experiments: (3a.) In some figures, the n value is provided, but not all. (3b.) Quantifications to accompany these observations is provided for Figures 4 and 5, but I could not locate it for Figure 6 or 7.

4. I recommend moving Figure 7, supplement 1 into the main paper as Figure 8. It will be very useful for readers looking at the main manuscript, and facilitate understanding of the authors' data and their interpretations of the mechanisms at play.

5. Typo: line 822, Figure 3 legend. ELISA is misspelled.

*Reviewer #2 (Recommendations for the authors):*

This paper is appropriate for publication in *eLife* if the following concerns are addressed in a revised manuscript.

Concerns:

1. The authors utilize single-cell mRNA sequencing to identify cell types in the adult zebrafish kidney, the mesonephros, and they classify a novel cell type as renal interstitial cells (RICs). However, other than col1a2, the orthologs of the markers used to distinguish this cell type (including fabp10a, col1a1b, col6a4a, *Mmp2*, cox2a, and epo) are not robustly expressed within the interstitial cells identified by single-cell mRNA sequencing in the adult mammalian nephron (see https://cello.shinyapps.io/kidneycellexplorer/). Furthermore, zfin does not list an ortholog of fabp10. Despite these differences, the cell type that the authors identified may in fact be interstitial cells and gene expression may vary between these species. Because the identification of this cell type is so important to this study, the list of genes expressed in these cells within the single-cell mRNA data could be used for comparison to provide more convincing evidence for this cell type and potentially identify differences that may be relevant in regeneration.

2. Proliferation should be quantitated in the same figures as the immunofluorescence marked by EdU staining.

3. Changes in β-catenin stability by immunostaining in Figures 6 and 7 and the related supplements are difficult to discern, especially since the nuclear β-catenin cannot be visualized. Because the signaling pool of β-catenin is sometimes difficult to visualize, a complementary experiment would add robustness to these experiments. Given that lhx1 cells were isolated by FACs in other experiments in the RPCs, β-catenin stability could be evaluated by this method or by using cell fractionation. Potentially quantitative imaging might help as well.

---

## [Author Response]

Essential revisions:Please address all the comments of the reviewers. A particular focus should be on the following points:1) In line with Reviewer #1/point 3 the number of samples analyzed should be indicated and the immunofluorescence should be quantified. In addition, please make sure that all the data are analyzed for statistical analysis when appropriate.

We have provided the experimental sample numbers (n) and quantified immunofluorescence results, as well as necessary statistics data according to reviewer 1’s suggestions (see response to reviewer 1/point 3).

2) In line with the comments of Reviewer #2/point 1 on the single-cell data, the clustering results need to be put into the context with the single-cell data from mouse and if possible from humans.

We have compared our zebrafish data with the single-cell RNA sequencing data of human and mouse kidneys. We have found that the main RIC marker genes, such as *col1a2* and proteoglycan *decorin* (*dcn),* are all expressed in zebrafish, mice and human. However, some RIC markers are expressed in zebrafish and human, but not in mice, while others are expressed in zebrafish and mice, and not in human, suggesting that gene expression may vary in RICs between these species. Although *fabp10a* is detectable only in zebrafish interstitial cells, *fabp3*, *fabp4*, and *fabp5* are highly expressed in mouse and human RICs (Ransick et al., 2019; Stewart et al., 2019; Young et al., 2018; https://cello.shinyapps.io/kidneycellexplorer/), revealing a concretive role of fatty acid-binding protein family members in marking RICs. Importantly, *fabp10a*-labbeled zebrafish interstitial cells exhibit the same morphology, distribution and biochemical characteristics as mammalian RICs. This cluster of cells is also enriched in extracellular matrix genes or genes that respond to lipids. Taken together, these findings concluded *fabp10a*-marked cell cluster as zebrafish RICs (see response to reviewer 2/point 1).

3) Reviewer #2/point 3 raises some concerns in respect to on the connection between PGE2 signaling and Wnt signaling. Please further clarify the interaction based on the suggestions of the review so that the general readership can appreciate its significance.

We understand the editor’s suggestions and reviewer’s concerns, and have performed new experiments to detect nuclear β-catenin and further clarify the interaction between PGE2 signaling and Wnt signaling. In particular, we have performed β-catenin immunostaining on FACS-sorted RPCs and found that nuclear β-catenin is significantly reduced in *cox2a*^-/-^ or *wnt4a^-/-^* RPCs compared to WT RPCs, and importantly, dmPGE2 treatment can restored the reduced β-catenin in these mutant RPCs (Figure 6—figure supplement 2). Furthermore, we have conducted fluorescence in situ hybridization experiments to detect the expression of the Wnt/β-catenin target gene *lef1*, and observed a reduction of *lef1* in *lhx1a^+^* RPC aggregates in *cox2a*^-/-^ and *wnt4a^-/-^* mutants, and dmPGE2 treatment rescued the decreased *lef1* expression (Figure 6—figure supplement 3). These findings, together with APC/β-catenin destruction complex data (Figure 6A, D–G, L), demonstrating strongly that PGE2–EP4 signaling regulates Wnt activity by regulating nuclear β-catenin levels for transcriptional controls during nephron regeneration. We have also quantified the immunostaining of p-S675-β-catenin and p-S9-GSK3β, and found p-S675-β-catenin and p-S9-GSK3β were decreased in the *lhx1a^+^* RPC aggregate in the absence of PGE2 (Figure 7X, Y), further corroborating that β-catenin stability is regulated by PGE2 signaling during renal regeneration (see response to reviewer 2/point 3).

Reviewer #1 (Recommendations for the authors):The manuscript is clearly written and was a joy to read. I have not read such a wonderful study in a very long time. I have a few questions and suggestions to improve the work but endorse it very strongly and enthusiastically for publication.1. Which part of the embryonic pronephros exhibits fabp10a expression at the 4-6 dpf stages (Figure 2, figure supplement 1A)? The relative position (proximal, distal, throughout the tubule) would be helpful to describe.

We have found that *fabp10a* is mainly enriched in the distal tubule (DT) and proximal straight tubule (PST) at early stages (Figure 2—figure supplement 1A). We have described the relative positions where *fabp10a* expressed in the manuscript and figure legends (lines 119–121, and 1044).

2. The manuscript would be well-served to cite and acknowledge the formative work of Renate Reimschuessel, who documented neonephrogenesis in fishes and established gentamicin for such studies, in addition to the team of Diep et al. 2011 that adopted this for zebrafish (line 148).

Thanks for the suggestion, and we have cited this paper in the manuscript (line 151).

3. In a number of figures, the authors show confocal imaging of representative kidneys stained by immunofluorescence. Two comments related to these experiments: (3a.) In some figures, the n value is provided, but not all. (3b.) Quantifications to accompany these observations is provided for Figures 4 and 5, but I could not locate it for Figure 6 or 7.

We have added n values in in all figures in their figure legends (lines 846, 851, 856, 882, 925, 1061, 1063, 1086, 1090, and 1106). We have also provided quantification analyses of the fluorescent pictures in *Figure 6* and *Figure 7* (Figure 6W; Figure 7X and Y).

4. I recommend moving Figure 7, supplement 1 into the main paper as Figure 8. It will be very useful for readers looking at the main manuscript, and facilitate understanding of the authors' data and their interpretations of the mechanisms at play.

As reviewer suggested, we have now moved and changed Figure 7—Figure supplement 1 as Figure 8.

5. Typo: line 822, Figure 3 legend. ELISA is misspelled.

We have spelling-checked the whole manuscript, and corrected these errors in line 844 and Figure 3 legend.

Reviewer #2 (Recommendations for the authors):This paper is appropriate for publication in eLife if the following concerns are addressed in a revised manuscript.Concerns:1. The authors utilize single-cell mRNA sequencing to identify cell types in the adult zebrafish kidney, the mesonephros, and they classify a novel cell type as renal interstitial cells (RICs). However, other than col1a2, the orthologs of the markers used to distinguish this cell type (including fabp10a, col1a1b, col6a4a, Mmp2, cox2a, and epo) are not robustly expressed within the interstitial cells identified by single-cell mRNA sequencing in the adult mammalian nephron (see https://cello.shinyapps.io/kidneycellexplorer/). Furthermore, zfin does not list an ortholog of fabp10. Despite these differences, the cell type that the authors identified may in fact be interstitial cells and gene expression may vary between these species. Because the identification of this cell type is so important to this study, the list of genes expressed in these cells within the single-cell mRNA data could be used for comparison to provide more convincing evidence for this cell type and potentially identify differences that may be relevant in regeneration.

Thanks for the thoughtful comments. By comparing our zebrafish data (Figure 1—figure supplement 1) with the single-cell RNA sequencing data of human and mouse kidneys (Ransick et al., 2019; Stewart et al., 2019; Young et al., 2018; https://cello.shinyapps.io/kidneycellexplorer/), we have found that the main RIC marker genes, such as *col1a2* and proteoglycan *decorin* (*dcn),* are all expressed in zebrafish, mice and human. However, *col1a1b, col6a2* and *Mmp2* are expressed in zebrafish and human RICs, but not in murine RICs; *elastin microfibril interfacer 1* (*emilin1a)* is expressed in zebrafish and mouse RICs, and not human interstitial cells, suggesting that gene expression may vary in RICs between these species. Notably, *fabp3*, *fabp4*, and *fabp5* are highly expressed in mouse and human RICs (Ransick et al., 2019; Stewart et al., 2019; Young et al., 2018; https://cello.shinyapps.io/kidneycellexplorer/), despite that *fabp10a* is detectable only in zebrafish interstitial cells, revealing a concretive role of fatty acid-binding protein family members in marking RICs. We have incorporated these findings in the Results and Discussion (lines 96–101, 412–415).

As the reviewer noticed that, in mammalian RICs, *Epo* and *Cox2* could not be identified by single-cell sequencing. However, Epo and Cox2 could be detectable in human and mouse RICs by immunostaining (*Kobayashi et al., 2016; Zhang et al., 2018*). The expressions of *epo and cox2* could also be detectable in FACS-sorted zebrafish RICs using RT-PCR (Figure 2D)*,* suggesting that Cox2 and Epo can be used as RIC markers between these species. Importantly, *fabp10a*-labbeled zebrafish interstitial cells exhibit the same morphology, distribution and biochemical characteristics as mammalian RICs (Figure 2). This cluster of cells is also enriched in extracellular matrix genes or genes that respond to lipids (Figure 1C). Taken together, these findings concluded *fabp10a*-marked cell cluster as zebrafish RICs.

2. Proliferation should be quantitated in the same figures as the immunofluorescence marked by EdU staining.

We have moved the quantification diagram of EdU from the supplement figures to the corresponding figures containing EdU immunofluorescence (Figure 4K; Figure 5P; and Figure 4—figure supplement 2G).

3. Changes in β-catenin stability by immunostaining in Figures 6 and 7 and the related supplements are difficult to discern, especially since the nuclear β-catenin cannot be visualized. Because the signaling pool of β-catenin is sometimes difficult to visualize, a complementary experiment would add robustness to these experiments. Given that lhx1 cells were isolated by FACs in other experiments in the RPCs, β-catenin stability could be evaluated by this method or by using cell fractionation. Potentially quantitative imaging might help as well.

We understand the reviewer’s concerns and suggestions, and have isolated RPCs by FACS in *cox2a*^-/-^ and *wnt4a^-/-^* mutants, as well as chemical-treated *cox2a*^-/-^ and *wnt4a^-/-^* mutants following AKI. The FACS-sorted RPCs are good enough to perform immunostaining of β-catenin but not cell fractionation. We have observed that nuclear β-catenin is significantly reduced in *cox2a*^-/-^ or *wnt4a^-/-^* RPCs compared to WT RPCs. Importantly, dmPGE2 treatment can restored the reduced β-catenin in these mutant cells (lines 353–357, Figure 6—figure supplement 2). Furthermore, our new fluorescence in situ hybridization (FISH) analyses showed that the reduced Wnt/β-catenin signaling in *cox2a*^-/-^and *wnt4a^-/-^* mutants caused a decrease in *lef1* expression in *lhx1a^+^* RPC aggregates, and dmPGE2 treatment rescued the decreased *lef1* expression (lines 357–363, Figure 6—figure supplement 3). These findings, together with APC/β-catenin destruction complex data (Figure 6A, D–G, L), demonstrating that PGE2–EP4 signaling regulates Wnt activity by regulating nuclear β-catenin levels for transcriptional controls during nephron regeneration. We have also quantified the immunostaining of p-S675-β-catenin and p-S9-GSK3β, both of which control β-catenin stability, and found that p-S675-β-catenin and p-S9-GSK3β were decreased in the *lhx1a^+^* RPC aggregate in the absence of PGE2 (Figure 7X, Y), further corroborating that β-catenin stability is regulated by PGE2 signaling during renal regeneration.

References

Kobayashi, H., Liu, Q., Binns, T. C., Urrutia, A. A., Davidoff, O., Kapitsinou, P. P.,... Haase, V. H. (2016). Distinct subpopulations of FOXD1 stroma-derived cells regulate renal erythropoietin. *J Clin Invest, 126*(5), 1926-1938. doi:10.1172/JCI83551

Ransick, A., Lindstrom, N. O., Liu, J., Zhu, Q., Guo, J. J., Alvarado, G. F.,... McMahon, A. P. (2019). Single-Cell Profiling Reveals Sex, Lineage, and Regional Diversity in the Mouse Kidney. *Dev Cell, 51*(3), 399-413 e397. doi:10.1016/j.devcel.2019.10.005

Stewart, B. J., Ferdinand, J. R., Young, M. D., Mitchell, T. J., Loudon, K. W., Riding, A. M.,... Clatworthy, M. R. (2019). Spatiotemporal immune zonation of the human kidney. *Science, 365*(6460), 1461-1466. doi:10.1126/science.aat5031

Young, M. D., Mitchell, T. J., Vieira Braga, F. A., Tran, M. G. B., Stewart, B. J., Ferdinand, J. R.,... Behjati, S. (2018). Single-cell transcriptomes from human kidneys reveal the cellular identity of renal tumors. *Science, 361*(6402), 594-599. doi:10.1126/science.aat1699

Zhang, M. Z., Wang, S., Wang, Y., Zhang, Y., Ming Hao, C., and Harris, R. C. (2018). Renal Medullary Interstitial COX-2 (Cyclooxygenase-2) Is Essential in Preventing Salt-Sensitive Hypertension and Maintaining Renal Inner Medulla/Papilla Structural Integrity. *Hypertension, 72*(5), 1172-1179. doi:10.1161/HYPERTENSIONAHA.118.11694